# COMPOSING ENTROPIC POLICIES USING DIVERGENCE CORRECTION

## ABSTRACT

Deep reinforcement learning (RL) algorithms have made great strides in recent years. An important remaining challenge is the ability to quickly transfer existing skills to novel tasks, and to combine existing skills with newly acquired ones. In domains where tasks are solved by composing skills this capacity holds the promise of dramatically reducing the data requirements of deep RL algorithms, and hence increasing their applicability. Recent work has studied ways of composing behaviors represented in the form of action-value functions. We analyze these methods to highlight their strengths and weaknesses, and point out situations where each of them is susceptible to poor performance. To perform this analysis we extend generalized policy improvement to the max-entropy framework and introduce a method for the practical implementation of successor features in continuous action spaces. Then we propose a novel approach which, in principle, recovers the optimal policy during transfer. This method works by explicitly learning the (discounted, future) divergence between policies. We study this approach in the tabular case and propose a scalable variant that is applicable in multi-dimensional continuous action spaces. We compare our approach with existing ones on a range of non-trivial continuous control problems with compositional structure, and demonstrate qualitatively better performance despite not requiring simultaneous observation of all task rewards.

## 1 INTRODUCTION

Reinforcement learning algorithms coupled with powerful function approximators have recently achieved a series of successes (Mnih et al., 2015; Silver et al., 2016; Lillicrap et al., 2015; Kalashnikov et al., 2018). Unfortunately, while being extremely powerful, deep reinforcement learning (DRL) algorithms often require a large number of interactions with the environment to achieve good results, partially because they are often applied "from scratch" rather than in settings where they can leverage existing experience. This reduces their applicability in domains where generating experience is expensive, or learning from scratch is challenging.

The data efficiency of DRL algorithms is affected by various factors and significant research effort has been directed at achieving improvements (e.g. Popov et al., 2017). At the same time the development of basic locomotor behavior in humans can, in fact, require large amounts of experience and practice (Adolph et al., 2012), and it can take significant effort and training to master complex, high-speed skills (Haith & Krakauer, 2013). Once such skills have been acquired, however, humans rapidly put them to work in new contexts and to solve new tasks, suggesting transfer learning as an important mechanism.

Transfer learning has been explored extensively in multiple fields of the machine learning community (see e.g. Weiss et al., 2016, for a recent review). In RL and robotics the transfer of knowledge from one task to another has been studied from a variety of angles.

For the purpose of this paper we are interested in methods that are suitable for transfer in the context of high-dimensional motor control problems. We further focus on model-free approaches, which are evident in human motor control (Haith & Krakauer, 2013), and have recently been used by a variety of scalable deep RL methods (e.g. Lillicrap et al., 2015; Mnih et al., 2015; Schulman et al., 2017; Kalashnikov et al., 2018).

Transfer may be especially valuable in domains where a small set of skills can be composed, in different combinations, to solve a variety of tasks. Different notions of compositionality have been considered in the RL and robotics literature. For instance, 'options' are associated with discrete units of behavior that can be sequenced, thus emphasizing composition in time (Precup et al., 1998). In this paper we are concerned with a rather distinct notion of compositionality, namely how to combine and blend potentially concurrent behaviors. This form of composition is particularly relevant in high-dimensional continuous action spaces, where it is possible to achieve more than one task simultaneously (e.g. walking somewhere while juggling).

One approach to this challenge is via the composition of task rewards. Specifically, we are interested in the following question: If we have previously solved a set of tasks with similar transition dynamics but different reward functions, how can we leverage this knowledge to solve new tasks which can be expressed as a convex combination of those rewards functions?

This question has recently been studied in two independent lines of work: by Barreto et al. (2017; 2018) in the context of successor feature (SF) representations used for Generalized Policy Improvement (GPI) with deterministic policies, and by Haarnoja et al. (2018a); van Niekerk et al. (2018) in the context of maximum entropy policies. These approaches operate in distinct frameworks but both achieve skill composition by combining the $Q$-functions associated with previously learned skills.

We clarify the relationship between the two approaches and show that both can perform well in some situations but achieve poor results in others, often in complementary ways. We introduce a novel method of behavior composition that that can consistently achieve good performance.

Our contributions are as follows:

1. We introduce succcessor features (SF) in the context of maximum entropy and extend the GPI theorem to this case (max-ent GPI).

2. We provide an analysis of when GPI, and compositional "optimism" (Haarnoja et al., 2018a) of entropy-regularized policies transfer. We construct both tabular and continuous action tasks where both fail to transfer well.

3. We propose a correction term – which we call Divergence Correction (DC)– based on the Rényi divergence between policies which allows us, in principle, to recover the optimal policy for transfer for any convex combination of rewards.

4. We demonstrate a practical implementation of these methods in continuous action spaces using adaptive importance sampling and compare the approaches introduced here: max-ent GPI and DC with optimism(Haarnoja et al., 2018a) and Conditional $Q$ functions (Schaul et al., 2015) in a variety of non-trivial continuous action transfer tasks.

## 2 BACKGROUND

### 2.1 MULTI-TASK RL

We consider Markov Decision Processes defined by the tuple $\mathcal{M}$ containing: a state space $\mathcal{S}$, action space $\mathcal{A}$, a start state distribution $p(s_1)$, a transition function $p(s_{t+1}|s_t, a_t)$, a discount $\gamma \in [0, 1)$ and a reward function $r(s_t, a_t, s_{t+1})$. The objective of RL is to find a policy $\pi(a|s) : \mathcal{S} \rightarrow \mathcal{P}(\mathcal{A})$ which maximises the discounted expected return from any state $J(\pi) = \mathbb{E}_{\pi,\mathcal{M}} \left[ \sum_{\tau=t}^{\infty} \gamma^{\tau-t} r_\tau \right]$ where the expected reward is dependent on the policy $\pi$ and the MDP $\mathcal{M}$.

We formalize transfer as in Barreto et al. (2017); Haarnoja et al. (2018a), as the desire to perform well across all tasks in a set $\mathcal{M} \in \mathcal{T}'$ after having learned policies for tasks $\mathcal{M} \in \mathcal{T}$, without additional experience. We assume that $\mathcal{T}$ and $\mathcal{T}'$ are related in two ways: all tasks share the same state transition function, and tasks in $\mathcal{T}'$ can be expressed as convex combinations of rewards associated with tasks in set $\mathcal{T}$. So if we write the reward functions for tasks in $\mathcal{T}$ as the vector $\phi = (r_1, r_2, \dots)$, tasks in $\mathcal{T}'$ can be expressed as $r_{\mathbf{w}} = \phi \cdot \mathbf{w}$.

We focus on combinations of two policies $r_b = br_i + (1 - b)r_j$ but the methods can be extended to more than two tasks. We refer to a transfer method as optimal, if it achieves optimal returns on tasks in $\mathcal{T}'$, using only experience on tasks $\mathcal{T}$.

## 2.2 Successor Features

Successor Features (SF) (Dayan, 1993) and Generalised Policy Improvement (GPI) (Barreto et al., 2017; 2018) provide a principled solution to transfer in the setting defined above. SF make the additional assumption that the reward feature $\phi$ is fully observable, that is, the agent has access to the rewards of all tasks in $\mathcal{T}$ but not $\mathcal{T}'$ during training on each individual task.

The key observation of SF representations is that linearity of the reward $r_{\mathbf{w}}$ with respect to the features $\phi$ implies the following decomposition of the value policy of $\pi$:

$$Q^\pi_{\mathbf{w}}(s_t, a_t) = \mathbb{E}_\pi \left[ \sum_{\tau=t}^\infty \gamma^{\tau-t} \phi_\tau \cdot \mathbf{w} | a_t \right] = \mathbb{E}_\pi \left[ \sum_{i=t}^\infty \gamma^{\tau-t} \phi_\tau | a_t \right] \cdot \mathbf{w} \equiv \psi^\pi(s_t, a_t) \cdot \mathbf{w}, \quad (1)$$

where $\psi^\pi$ is the expected discounted sum of features $\phi$ induced by policy $\pi$. This decomposition allows us to compute the action-value for $\pi$ on any task $\mathbf{w}$ by learning $\psi^\pi$.

If we have a set of policies $\pi_1, \pi_2, ..., \pi_n$ indexed by $i$, SF and GPI provide a principled approach to transfer on task $\pi_{\mathbf{w}}$. Namely, we act according to the deterministic GPI policy $\pi^{GPI}_{\mathbf{w}}(s_t) \equiv \arg\max_{a_t} Q^{GPI}_{\mathbf{w}}(s_t, a_t))$ where

$$Q^{GPI}_{\mathbf{w}}(s_t, a_t) \equiv \max_i Q^{\pi_i}_{\mathbf{w}}(s_t, a_t) = \max_i \psi^{\pi_i}(s, a) \cdot \mathbf{w} \quad (2)$$

The GPI theorem guarantees the GPI policy has a return at least as good as any component policy, that is, $V^{\pi^{GPI}_{\mathbf{w}}}(s) \geq \max_i V^{\pi_i}_{\mathbf{w}}(s) \, \forall s \in \mathcal{S}$.

## 2.3 Maximum Entropy RL

The maximum entropy (max-ent) RL objective augments the reward to favor entropic solutions

$$J(\pi) = \mathbb{E}_{\pi, \mathcal{M}} \left[ \sum_{i=\tau}^\infty \gamma^{\tau-t}(r_\tau + \alpha H[\pi(\cdot|s_\tau)]) \right] \quad (3)$$

where $\alpha$ is a parameter that determines the relative importance of the entropy term.

This objective has been considered in a number of works including Kappen (2005); Todorov (2009); Haarnoja et al. (2017; 2018a); Ziebart et al. (2008); Fox et al. (2015).

We define the action-value $Q^\pi$ associated with eq. 3 as

$$Q^\pi(s_t, a_t) \equiv r_t + \mathbb{E}_\pi \left[ \sum_{\tau=t+1}^\infty \gamma^{\tau-t}(r_\tau + \alpha H[\pi(\cdot|s_\tau)]) \right] \quad (4)$$

(notice $Q^\pi(s_t, a_t)$ does not include any entropy terms for the state $s_t$). Soft Q iteration

$$Q(s_t, a_t) \leftarrow r(s_t, a_t, s_{t+1}) + \gamma \mathbb{E}_{p(s_{t+1}|s_t, a_t)} [V(s_{t+1})] \quad (5)$$

$$V(s_t) \leftarrow \mathbb{E}_\pi [Q(s_t, a_t)] + \alpha H[\pi(\cdot|s_t)] = \alpha \log \int_{\mathcal{A}} \exp(\frac{1}{\alpha} Q(s_t, a_t)) da \equiv \alpha \log Z(s_t) \quad (6)$$

where $\pi(a_t|s_t) \propto \exp(\frac{1}{\alpha} Q(s_t, a_t))$ converges to the optimal policy with standard assumptions (Haarnoja et al., 2017).

## 3 Composing Policies in Max-Ent Reinforcement Learning

In this section we present two novel approaches for max-ent transfer learning. In section 4 we then outline a practical method for making use of these results.

## 3.1 Max-Ent Successor Features and Generalized Policy Improvement

We introduce max-ent SF, which provide a practical method for computing the value of a maximum entropy policy under any convex combination of rewards. We then show the GPI theorem (Barreto et al., 2017) holds for maximum entropy policies.

We define the action-dependent SF to include the entropy of the policy, excluding the current state, analogous to the max-entropy definition of $Q^\pi$ in (4):

$$\psi^\pi(s_t, a_t) \equiv \phi_t + \mathbb{E}_\pi \left[ \sum_{\tau=i+1}^\infty \gamma^{\tau-t}(\phi_\tau + \alpha \mathbf{1} \cdot H[\pi(\cdot|s)]) \right] = \phi_t + \gamma \mathbb{E}_{p(s_{t+1}|s_t, a_t)} [\Upsilon(s_{t+1})] \quad (7)$$

where $\mathbf{1}$ is a vector of ones of the same dimensionality as $\phi$ and we define the state-dependent successor features as the expected $\boldsymbol{\psi}^\pi$ in analogy with $V^\pi(s)$:

$$\mathbf{\Upsilon}^\pi(s) \equiv \mathbb{E}_{a \sim \pi(\cdot|s)}\left[\boldsymbol{\psi}^\pi(s,a)\right] + \alpha\mathbf{1} \cdot H[\pi(\cdot|s)]. \tag{8}$$

The max-entropy action-value of $\pi$ for any convex combination of rewards $\mathbf{w}$ is then given by $Q_\mathbf{w}^\pi(s,a) = \boldsymbol{\psi}^\pi(s,a) \cdot \mathbf{w}$. Max-ent SF allow us to estimate the action-value of previous policies on a new task. We show that, as in the deterministic case, there is a principled way to combine multiple policies using their action-values on task $\mathbf{w}$.

**Theorem 3.1 (Max-Ent Generalized Policy Improvement)** *Let $\pi_1, \pi_2, ..., \pi_n$ be $n$ policies with $\alpha$-max-ent action-value functions $Q^1, Q^2, ..., Q^n$ and value functions $V^1, V^2, ..., V^n$. Define*

$$\pi(a|s) \propto \exp\left(\tfrac{1}{\alpha}\max_i Q^i(s,a)\right).$$

*Then,*

$$Q^\pi(s,a) \geq \max_i Q^i(s,a) \text{ for all } s \in \mathcal{S} \text{ and all } a \in \mathcal{A}, \tag{9}$$

$$V^\pi(s) \geq \max_i V^i(s) \text{ for all } s \in \mathcal{S}, \tag{10}$$

*where $Q^\pi(s,a)$ and $V^\pi(s)$ are the $\alpha$-max-ent action-value and value function respectively of $\pi$.*

Proof: See appendix A.1. In our setup, we learn $\boldsymbol{\psi}^{\pi_i}(s,a)$, the SFs of policies $\pi_i$ for each task in $\mathcal{T}$, we define the max-ent GPI policy for task $\mathbf{w} \in \mathcal{T}'$ as $\pi_\mathbf{w}^{GPI}(a|s) \propto \exp(\frac{1}{\alpha}\max_i Q_\mathbf{w}^{\pi_i}(s,a)) = \exp(\frac{1}{\alpha}\max_i \boldsymbol{\psi}^{\pi_i}(s,a) \cdot \mathbf{w})$.

## 3.2 Divergence Correction (DC)

Haarnoja et al. (2018a) introduced a simple approach to policy composition by estimating the action-value for the transfer task $r_b = br_i + (1-b)r_j$ from the optimal action-values of the component tasks $Q^i$ and $Q^j$

$$Q_b^{Opt}(s,a) \equiv bQ^i(s,a) + (1-b)Q^j(s,a). \tag{11}$$

When using Boltzmann policies defined by $Q$, the resulting policy, $\pi_b^{Opt}(a|s) \propto \exp(\frac{1}{\alpha}Q_b^{Opt}(s,a))$, is the product distribution of the two component policies. We refer to $\pi_b^{Opt}$ as the compositionally "optimistic" (CO) policy, as it acts according to the optimistic assumption that the optimal returns of $Q^i$ and $Q^j$ will be, simultaneously, achievable[1].

Both max-ent GPI we presented above, and CO can, in different ways, fail to transfer well in some situations (see fig. 1 for some examples in tabular case). Neither approach consistently performs optimally during transfer, even if all component terms are known exactly. We desire a solution for transfer that, in principle, can perform optimally.

Here we show, at the cost of learning a function conditional on the task weightings $b$, it is in principle possible to recover the optimal policy for the transfer tasks, without direct experience on those tasks, by correcting for the compositional optimism bias in $Q_b^{Opt}$. For simplicity, as in Haarnoja et al. (2018a), we restrict this to the case with only 2 tasks, but it can be extended to multiple tasks.

The correction term for CO uses a property noted, but not exploited in Haarnoja et al. (2018a). The bias in $Q^{Opt}$ is related to the the discounted sum of Rényi divergences of the two component policies. Intuitively, if the two policies result in trajectories with low divergence between the policies in each state, the CO assumption that both policies can achieve good returns is approximately correct. When the divergences are large, the CO assumption is being overly optimistic and the correction term will be large.

**Theorem 3.2 (DC Optimality)** *Let $\pi_i, \pi_j$ be $\alpha$ max-ent optimal policies for tasks with rewards $r_i$ and $r_j$ with max-ent action-value functions $Q^i, Q^j$. Define $C_b^\infty(s_t, a_t)$ as the fixed point of*

$$C_b^{(k+1)}(s_t, a_t) = -\alpha\gamma\mathbb{E}_{p(s_{t+1}|s_t,a_t)}\left[\log \int_\mathcal{A} \pi_i(a_{t+1}|s_{t+1})^b \pi_j(a_{t+1}|s_{t+1})^{(1-b)}\exp(-\tfrac{1}{\alpha}C_b^{(k)}(s_{t+1},a_{t+1}))da_{t+1}\right]$$

---

[1]Compositional optimism is not the same as optimism under uncertainty, often used in RL for exploration.

*Given the conditions for Soft Q convergence, the max-ent optimal $Q_b^*(s, a)$ for $r_b = br_i + (1 - b)r_j$ is*

$$Q_b^*(s, a) = bQ^i(s, a) + (1 - b)Q^j(s, a) - C_b^\infty(s, a) \; \forall s \in \mathcal{S}, a \in \mathcal{A}, b \in [0, 1].$$

Proof: See appendix A.2. We call this Divergence Correction (DC) as the quantity $C_b^\infty$ is related to the Rényi divergence between policies (see appendix A.2 for details). Learning $C_b^\infty$ does not require any additional information (in principle) than that required to learn policies $\pi_i$ and $\pi_j$. Unlike with SF, it is not necessary to observe other task features while training the policies. On the other hand, unlike with GPI, which can be used to naturally combine any number of tasks with arbitrary weight vectors $\mathbf{w}$, in order to apply DC one must estimate $C_b^\infty(s, a)$ for all values of $b$. so the complexity of learning $C^\infty$ increases significantly if more than 2 tasks are combined.

Supplementary Table 1 provides a comparison on the properties of the methods we consider here. We also compare with simply learning a conditional $Q$ function $Q(s, a|b)$ (CondQ) (e.g. Schaul et al., 2015; Andrychowicz et al., 2017). As with GPI, this requires observing the full set of task features $\phi$, in order to compute $r_b$ for arbitrary $b$.

In this section we have introduced two new theoretical approaches to max-ent transfer composition: max-ent GPI and DC. We have shown how these are related to relevant prior methods. In the next section we address the question of how to practically learn and sample with these approaches in continuous action spaces.

## 4 ADAPTIVE IMPORTANCE SAMPLING FOR BOLTZMAN POLICIES ALGORITHM

The control of robotic systems with high-dimensional continuous action spaces is a promising use case for the ideas presented in this paper. Such control problems may allow for multiple solutions, and can exhibit exploitable compositional structure. Unfortunately, learning and sampling of general Boltzmann policies defined over continuous action spaces is challenging. While this can be mitigated by learning a parametric sampling distribution, during transfer we want to sample from the Boltzmann policy associated with a newly synthesized action-value function without having to learn such an approximation first. To address this issue we introduce Adaptive Importance Sampling for Boltzmann Policies (AISBP), a method which provides a practical solution to this challenge.

In the following we parametrise all functions with neural nets (denoting parameters by the subscript $\theta$), including the soft action-value for reward $i$: $Q_{\theta_Q}^i(s, a)$; the associated soft value function $V_{\theta_V}^i(s)$ and a proposal distribution $q_{\theta_q}^i(a|s)$, the role of which we explain below. We use an off-policy algorithm, so that experience generated by training on policy $i$ can be used to improve policy $j$. This is especially important since our analysis requires the action-value $Q^i(s, a)$ to be known in all states. This is less likely to be the case for a on on-policy algorithm, that only updates $Q^i$ using trajectories generated by policy $\pi^i$. During training experience generated by all tasks are stored in a replay buffer $R$, and mini-batches are sampled uniformly and used to update all function approximators. Soft Q iteration (see eq. 4) is used to learn $Q^i$ and $V^i$. These updates are, in principle, straightforward using transitions sampled from the replay buffer.

Sampling from the Boltzmann policy defined by $Q_{\theta_Q}^i$, $\pi^i(a|s) \propto \exp \frac{1}{\alpha} Q_{\theta_Q}^i(s, a)$ is challenging as is estimating the partition function (the $\log$ of which is also the value, c.f. Eq. 6). One approach is to fit an expressible, tractable sampler, such as a stochastic neural network to approximate $\pi^i$ (e.g. Haarnoja et al., 2018a). This approach works well when learning a single policy. However, during transfer this may require learning a new sampler for each new value composition. AISBP instead uses importance sampling to sample $\pi$ and estimate the partition function. The scalability of this approach is improved by using using a learned proposal distribution $q_{\theta_q}(a|s)$, and by observing that modern architectures allow for efficient batch computation of a large number of importance samples. To facilitate transfer we restrict the parametric form of the proposals to mixtures of (truncated) Normal distributions. The well-known result that the product of Normal distributions can be computed in closed-form then allows us to construct effective compositional proposals during transfer.

More formally, for each policy in $\mathcal{T}$ we learn an action-value $Q_{\theta_Q}^i(s, a)$, and value $V_{\theta_V}^i(s)$ network, and a proposal distribution $q_{\theta_q}^i(a|s)$ (we drop the task index $i$ here when writing the losses for nota-

tional clarify, and write the losses for a single policy). The proposal distribution is a mixture of $M$ truncated Normal distributions $\mathcal{N}_T$, truncated to the square $a \in [-1, 1)^n$ with diagonal covariances

$$q_{\theta_q}(a|s) = \frac{1}{M} \sum_{m=1}^{M} \mathcal{N}_T(a; \mu_{\theta_q}^m(s), \sigma_{\theta_q}^m(s), -1, 1) \tag{12}$$

The proposal distribution is optimized by minimizing the forward KL divergence with the Boltzmann policy $\pi(a|s) \propto \exp \frac{1}{\alpha} Q_{\theta_Q}(s, a)$. This KL is "zero avoiding" and over-estimates the support of $\pi$ (Murphy, 2012) which is desirable for a proposal distribution (Gu et al., 2015),

$$\mathcal{L}(\theta_q) = \mathbb{E}_R \left[ \mathbb{E}_{a \sim \pi(\cdot|s)} [\log \pi(a|s_t) - \log q_{\theta_q}(a|s_t)] \right] \tag{13}$$

where the expectation is over the replay buffer state density.

The inner expectation in the proposal loss itself requires sampling from $\pi$. We approximate his expectation by self-normalized importance sampling and use a target proposal distribution $p(a_t|s_t)$ which is a mixture distribution consisting of the proposals for all policies along with a uniform distribution. For batchsize $B$ and $N$ proposal samples the estimator of the proposal loss is then

$$\mathcal{L}(\theta_q) \approx -\frac{1}{B} \sum_{k=1}^{B} \sum_{l=1}^{N} w_{kl} \log q_{\theta_q}(a|s_t); \quad w_{kl}' = \frac{\frac{1}{\alpha}(Q_{\theta_Q}(s_k, a_{kl}))}{p(a_{kl}|s_k)}; \quad w_{kl} = \frac{w_{kl}'}{\sum_{m=1}^{N} w_{km}'}. \tag{14}$$

The value function loss is defined as the L2 error on the Soft Q estimate of value

$$\mathcal{L}(\theta_V) = \mathbb{E}_R \left[ \frac{1}{2} \left( V_{\theta_V}(s_t) - \alpha \log \int_{\mathcal{A}} \exp(\frac{1}{\alpha} Q_{\theta_Q}(s_t, a)) da \right)^2 \right] \tag{15}$$

which is estimated using importance sampling to compute the integral.

$$\mathcal{L}(\theta_V) \approx \frac{1}{2B} \sum_{l=1}^{B} \left( V_{\theta_V}(s_l) - \alpha \log Z \right)^2; \qquad Z = \left[ \frac{1}{N} \sum_{k=1}^{N} \frac{\exp(\frac{1}{\alpha} Q_{\theta_Q}(s_l, a_{lk}))}{q_{\theta_q}(a_{lk}|s_l)} \right]. \tag{16}$$

This introduces bias due to the finite-sample approximation of the expectation inside the (concave) log. In practice we found this estimator sufficiently accurate, provided the proposal distribution was close to $\pi$. We also use importance sampling to sample from $\pi$ while acting.

The action-value loss is just the L2 norm with the Soft Q target:

$$\mathcal{L}(\theta_Q) = \mathbb{E}_R \left[ \frac{1}{2} (Q_{\theta_Q}(s_t, a_t) - (r(s_t, a_t, s_{t+1}) + \gamma V_{\theta_V'}(s_{t+1})))^2 \right]. \tag{17}$$

To improve stability we employ target networks for the value $V_{\theta_{V'}}$ and proposal $q_{\theta_q'}$ networks (Mnih et al., 2015; Lillicrap et al., 2015) We also parameterize $Q$ as an advantage $Q_{\theta_Q}(s, a) = V_{\theta_V}(s) + A_{\theta_A}(s, a)$ (Baird, 1994; Wang et al., 2015; Harmon et al., 1995) which is more stable when the advantage is small compared with the value. The full algorithm is give in Algorithm Box 1 and more details are provided in appendix C.

## 4.1 IMPORTANCE SAMPLED MAX-ENT GPI

The same importance sampling approach can also be used to estimate max-ent SF. Max-ent GPI requires us to learn the expected (maximum entropy) features $\psi_i$ for each policy $\pi_i$, in order to estimate its (entropic) value under a new convex combination task $\mathbf{w}$. This requires that experience tuple in the replay contain the full feature vector $\phi$, rather than just the reward for the policy which generated the experience $r_i$. Given this information $\psi_{\theta_\psi}$ and $\Upsilon_{\theta_\Upsilon}$ can be learned with analogous updates to $V$ and $Q$, which again requires importance sampling to estimate $\Upsilon$.

As with $V_{\theta_V}$, we use a target network for $\Upsilon_{\theta_\Upsilon'}$ and advantage parametrization. We found that, because these updates when using experience shared between tasks is far off-policy, it is necessary to have a longer target update period than for $V$. Full details are of the losses and samplers are in appendix C.

---

**Algorithm 1** AISBP training algorithm

---

Initialize proposal network $\theta_q$, value network parameters $\theta_V$ and action-value network parameters $\theta_Q$ and replay $R$

**while** training **do**                                              ▷ in parallel on each actor
 Obtain parameters $\theta$ from learner
 Sample task $i \sim \mathcal{T}$
 Roll out episode using $q^i_{\theta_q}$ to importance sample $\pi_i(a|s) \propto \exp \frac{1}{\alpha} Q^i_{\theta_Q}(s, a)$
 Add experience to replay $R$
**end while**
**while** training **do**                                              ▷ in parallel on the learner
 Sample SARS tuple from $R$
 Improve $\mathcal{L}(\theta_q), \mathcal{L}(\theta_V), \mathcal{L}(\theta_Q)$
 Improve additional losses for transfer $\mathcal{L}(\theta_{\Upsilon}), \mathcal{L}(\theta_{\psi}), \mathcal{L}(\theta_C), \mathcal{L}(\theta_{V_b}) \mathcal{L}(\theta_{Q_b})$,
 **if** target update period **then**
  Update target network parameters $\theta_{V'} \leftarrow \theta_V, \theta_{q'} \leftarrow \theta_q, \theta_{\Upsilon'} \leftarrow \theta_{\Upsilon}, \theta_{V'_b} \leftarrow \theta_{V_b}$
 **end if**
**end while**

---

## 4.2 DIVERGENCE CORRECTION

All that is required for transfer using compositional optimism (eq. 11, Haarnoja et al. (2018a)) is the max-ent action values of each task, so no additional training is required beyond the base policies. In section 3.2 we have shown that if we can learn the fixed point of $C_b^\infty(s, a)$ we can correct this compositional optimism and recover the optimal action-value $Q_b^*(s, a)$.

We exploit the recursive relationship in $C_b^\infty(s, a)$ to fit a neural net $C_{\theta_C}(s, a, b)$ with a TD(0) estimator. This requires learning a conditional estimator for any value of $b$, so as to support arbitrary task combinations. Fortunately, since $C_b^\infty$ depends only on the policies and transition function it is possible to learn an estimator $C_b^\infty$ for different values of $b$ by sampling $b$ during each update. As before, we use target networks and an advantage parametrization for $C_{\theta_C}(s, a, b)$

We learn $C_b^\infty$ as $C_{\theta_C}(s, a, b)$, for each pair of policies $\pi_i, \pi_j$ resulting in the loss

$$\mathcal{L}(\theta_C) = \mathbb{E}_{s \sim R, b \sim U(0,1)}[\frac{1}{2}(C_{\theta_C}(s, a, b) + \alpha\gamma\mathbb{E}_{p(s'|s,a)}[\log \int_{\mathcal{A}} \exp(b \log \pi_i(a'|s')+ \tag{18}$$

$$(1 - b)\pi_j(a'|s') - \frac{1}{\alpha}C_{\theta_{C'}}(s', a', b))da'])^2].$$

As with other integrals of the action space, we approximate this loss using importance sampling to estimate the integral. Note that, unlike GPI and CondQ (next section), learning $C_b^\infty$ does not require observing $\phi$ while training.

We also considered a heuristic approach where we learned $C$ only for $b = \frac{1}{2}$ (this is typically approximately the largest divergence). This avoids the complexity of a conditional estimator and we estimate $C_b^\infty$ as $\hat{C}_b^\infty(s, a) \sim 4b(1 - b)C_{1/2}^\infty(s, a)$. This heuristic, we denote DC-Cheap, can be motivated by considering Gaussian policies with similar variance (see appendix D) The max-ent GPI bound can be used to correct for over-estimates of the heuristic $C_b^\infty$, $Q^{DC-Cheap+GPI}(s, a) = \max(Q^{OPT}(s, a) - \hat{C}_b^\infty(s, a), Q^{GPI}(s, a))$.

## 4.3 COND Q

As a baseline, we directly learn a conditional $Q$ function using a similar approach to DC of sampling $b$ each update $Q(s, a, b)$ (Schaul et al., 2015). This, like GPI but unlike DC, requires observing $\phi$ during training so the reward on task $b$ can be estimated. We provide the full details in appendix C.

## 4.4 SAMPLING COMPOSITIONAL POLICIES

During transfer we would like to be able to sample from the Boltzmann policy defined by our estimate of the transfer action-value $Q_b$ (the estimate is computed using the methods we enumerated

above) without having to, offline, learn a new proposal or sampling distribution first (which is the approach employed by Haarnoja et al. (2018a)).

As outlined earlier, we chose the proposal distributions so that the product of proposals is tractable, meaning we can sample from $q_b^{ij}(a|s) \propto (q_{\theta_q}^i(a|s))^b (q_\theta^j(a|s))^{(1-b)}$. This is a good proposal distribution when the CO bias is low, since $Q_b^{Opt}$ defines a Boltzmann policy which is the product of the base policies[2] However, when $C_b^\infty(s,a)$ is large, meaning the CO bias is large, $q^{ij}$ may not be a good proposal, as we show in the experiments. In this case none of the existing proposal distributions may be a good fit. Therefore we sample from a mixture distribution of all policies, all policy products and the uniform distribution.

$$p_b(a|s) \equiv \tfrac{1}{4}(q_{\theta_q}^i(a|s) + q_{\theta_q}^j(a|s) + q_b^{ij}(a|s) + \tfrac{1}{\mathcal{V}^A}) \qquad (19)$$

where $\mathcal{V}^A$ is the volume of the action space. Empirically, we find this is sufficient to result in good performance during transfer. The algorithm for transfer is given in supplementary algorithm 2.

## 5 EXPERIMENTS

### 5.1 DISCRETE, TABULAR ENVIRONMENT

We first consider some illustrative tabular cases of compositional transfer. These highlight situations in which GPI and CO transfer can perform poorly (Figure 1). As expected, we find that GPI performs well when the optimal transfer policy is close to one of the existing policies; CO performs well when both subtask policies are compatible. The task we refer to as "tricky" is illustrative of in which the optimal policy for the transfer task does not resemble either existing policy: In the grid world non-overlapping rewards for each task are provided in one corner of the grid world, while lower value overlapping rewards are provided in the other corner (cf. Fig. 1). As a consequence both GPI and CO perform poorly while DC performs well in all cases.

### 5.2 CONTINUOUS ACTION SPACES

We next compare the different approaches in more challenging continuous control tasks. We train max-ent policies to solve individual tasks using the importance sampling approach from section 4 and then assess transfer on convex combinations of the rewards. All approaches use the same experience and proposal distribution.

Figure 2 examines the transfer policies in detail in a simple point-mass task and shows how the estimated $C_b^\infty$ corrects the CO $Q^{Opt}$ and dramatically changes the policy.

We then examine conceptually similar tasks in more difficult domains: a 5 DOF planar manipulator reaching task (figure 3), 3 DOF jumping ball and 8 DOF ant (figure 4). We see that DC recovers a qualitatively better policy in all cases. The performance of GPI depends noticeably on the choice of $\alpha$. DC-Cheap, which is a simpler heuristic, performs almost as well as DC in the tasks we consider except for the point mass task. When bounded by GPI (DC-Cheap+GPI) it performs well for the point mass task as well, suggesting simple approximations of $C_b^\infty$ may be sufficient in some cases.[3]

We focussed on "tricky" tasks as they are challenging form of transfer. In general, we would expect DC to perform well in most situations where OC performs well, since in this case the correction term $C_b^\infty$ that DC must learn is inconsequential (OC is equivalent to assuming $C_b^\infty = 0$). Supplementary figure 5 demonstrates on a task with non-composible solutions (i.e. $C_b^\infty$ is large and potentially challenging to learn), DC continues to perform as well as GPI, slightly better than CondQ, and as expected, OC performs poorly.

## 6 DISCUSSION

We have presented two approaches to transfer learning via convex combinations of rewards in the maximum entropy framework: max-ent GPI and DC. We have shown that, under standard assump-

---

[2] $\pi_b^{Opt}(a|s) \propto \exp \tfrac{1}{\alpha} Q^{Opt}(s,a) = \exp(\tfrac{1}{\alpha}(Q^1(s,a) + Q^2(s,a)) = \pi_1(a|s)\pi_2(a|s)$.

[3] We provide videos of the more interesting tasks at `https://tinyurl.com/yaplfwaq`.

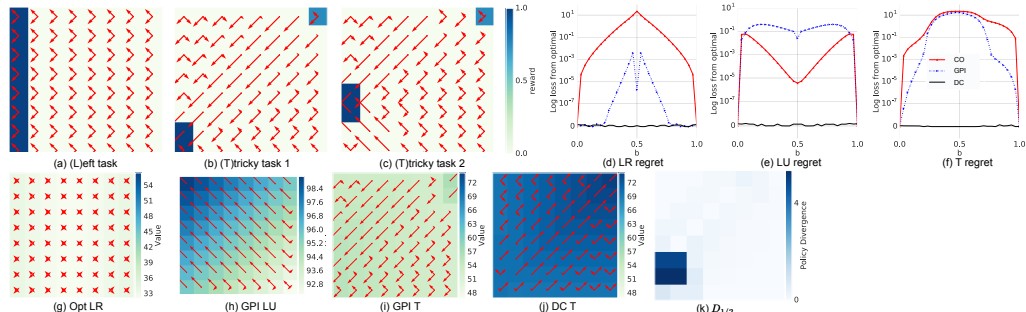

Figure 1: **Policy composition in the tabular case**. All tasks are in an infinite-horizon tabular 8x8 world. The action space is the 4 diagonal movements (actions at the boundary transition back to the same state) (**a-c**) shows 3 reward functions (color indicates reward, dark blue $r = +1$, light blue $r = 0.75$). The arrows indicate the action likelihoods for the max-ent optimal policy for each task. (**d-f**) The log regret for the max-ent returns for 3 qualitatively distinct compositional tasks $r_b = br_i + (1-b)r_j$, using different approaches to transfer from the base policies. The compositional tasks we consider are left-right (LR), left-up (LU) and the "tricky" tasks (T).
(**d**) GPI performs well when the subtasks are incompatible, meaning the optimal policy is near one of the component policies. (**g**) CO performs poorly in these situations, resulting in indecision about which subtask to commit to.
(**e**) Conversely, when the subpolicies are compatible, such as on the LU task, CO transfers well while the GPI policy (**h**) does not consistently take advantage of the compatibility of the two tasks to simultaneously achieve both subgoals.
(**f**) Neither GPI nor CO policies (**i** shows the GPI policy, but CO is similar) perform well when the optimal transfer policy is dissimilar to either existing task policy. The two tricky task policies are compatible in many states but have a high-divergence in the bottom-left corner since the rewards are non-overlapping there (**k**), thus the optimal policy on the composed task is to move to the top right corner where there are overlapping rewards. By learning, and correcting for, this future divergence between policies, DC results in optimal policies for all task combinations including tricky (**j**).

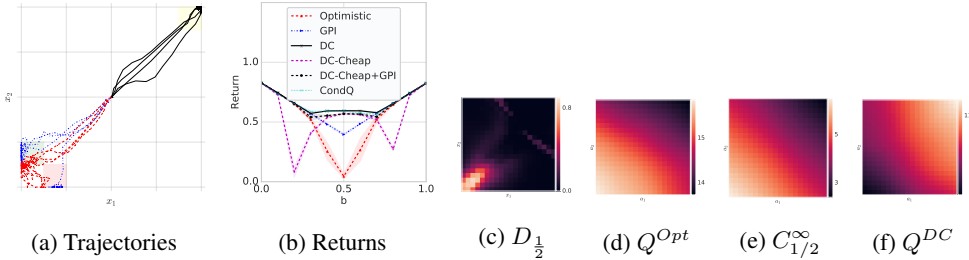

Figure 2: **Tricky point mass**. The continuous "tricky" task with a simple 2-D velocity controlled pointed mass. (**a**) Environment and example trajectories. The rewards are $(r_1 = 1, r_2 = 0)$, $(0, 1)$ and $(0.75, 0.75)$ for the green, red and yellow squares. Lines show sampled trajectories (starting in the center) for the compositional task $r_{1/2}$ with CO (red), GPI (blue) and DC (black). Only DC, DC heuristics and CondQ (not shown) find the optimal transfer policy of navigating to yellow reward area for the joint task which is the optimal solution for the compositional task. (**b**) The returns for each transfer method. DC and CondQ methods recover significantly better performance than GPI, and the CO policy performs poorly. (**c**) The Rényi divergence of the two base policies as a function of position: the two policies are compatible except near the bottom left corner where the rewards are non-overlapping. (**d**) $Q^{Opt}$ at the center position for the combined task. As both policies prefer moving left and down, most of the energy is on these actions. (**e**) However, the future divergence $C_{1/2}^{\infty}$ under these actions is high, which results in the (**f**) DC differing significantly from CO.

tions, the max-ent GPI policy performs at least as well as its component policies, and that DC recovers the optimal transfer policy. Todorov (2009) and (Saxe et al., 2017; van Niekerk et al., 2018) previously considered optimal composition of max-ent policies. However, these approaches

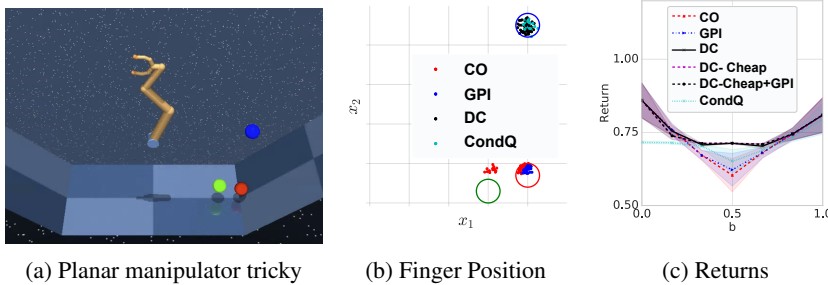

(a) Planar manipulator tricky     (b) Finger Position     (c) Returns

Figure 3: **"Tricky" task with planar manipulator**. The "tricky" tasks with a 5D torque-controlled planar manipulator. The training tasks consists of (mutually exclusive) rewards of $(1, 0), (0, 1)$ when the finger is at the green and red targets respectively and reward $(0.75, 0.75)$ at the blue target. (**b**) Finger position at the end of the trajectories starting from randomly sampled start states) for the transfer task with circles indicating the rewards. DC and CondQ trajectories reach towards the blue target (the optimal solution) while CO and GPI trajectories primarily reach towards one of the suboptimal partial solutions. (**c**) The returns on the transfer tasks (shaded bars show SEM, 5 seeds).

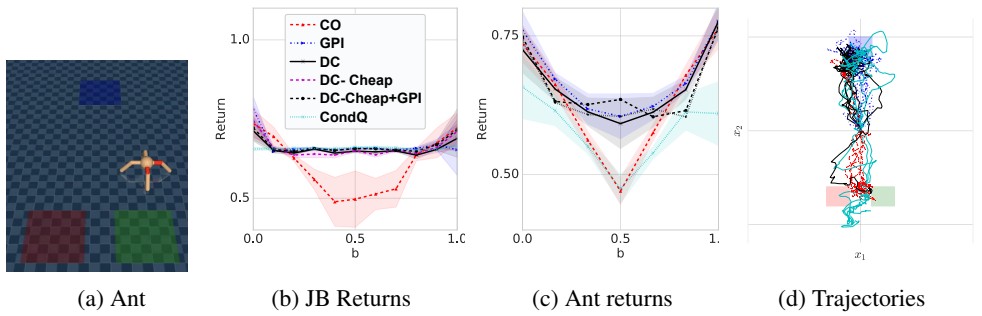

(a) Ant     (b) JB Returns     (c) Ant returns     (d) Trajectories

Figure 4: **"Tricky" task with mobile bodies.** "Tricky" task with two bodies: a 3 DOF jumping ball (supplementary figure 6) and (**a**) 8 DOF ant (both torque controlled). The task has rewards $(1, 0), (0, 1)$ in the green and red boxes respectively and $(0.75, 0.75)$ in the blue square. (**b-c**) Returns for both walkers when started in the center position. CO approach does not recover the optimal policy for the compositional task while the other approaches largely do, although CondQ does not learn a good policy on the ant (shaded bars show SEM, 3 seeds for jumping ball, 5 seeds for ant). (**e**) Sampled trajectories of the ant on the transfer task starting from a neutral position for $b = \frac{1}{2}$. GPI and DC consistently go to the blue square (optimal), CondQ and CO do not.

require stronger assumptions than max-ent SF or DC, namely that reward states are absorbing and that the joint reward is restricted to the softmax of the component rewards (soft OR). By contrast, DC does not restrict the class of MDPs and *learns* how compatible policies are, allowing approximate recovery of optimal transfer policies both when the component rewards are jointly achievable (AND), and when only one sub-goal can be achieved (OR).

We have compared our methods with conditional action-value functions (CondQ) (Schaul et al., 2015, e.g.) and optimistic policy combination (Haarnoja et al., 2018a). Further, we have presented AISBP, a practical algorithm for training DC and max-ent GPI models in continuous action spaces using adaptive importance sampling. We have compared these approaches, along with heuristic approximations of DC, and demonstrated that DC recovers an approximately optimal policy during transfer across a variety of high-dimensional control tasks. Empirically we have found CondQ may be harder to learn than DC, and it requires additional observation of $\phi$ during training.

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

# A  PROOFS

## A.1  MAX-ENT GENERALIZED POLICY IMPROVEMENT

**Theorem 3.1 (Max-Ent Generalized Policy Improvement)** *Let $\pi_1, \pi_2, ..., \pi_n$ be $n$ policies with $\alpha$-max-ent action-value functions $Q^1, Q^2, ..., Q^n$ and value functions $V^1, V^2, ..., V^n$. Define*

$$\pi(a|s) \propto \exp\left(\tfrac{1}{\alpha} \max_i Q^i(s,a)\right).$$

*Then,*

$$Q^\pi(s,a) \geq \max_i Q^i(s,a) \text{ for all } s \in \mathcal{S} \text{ and all } a \in \mathcal{A}, \tag{9}$$

$$V^\pi(s) \geq \max_i V^i(s) \text{ for all } s \in \mathcal{S}, \tag{10}$$

*where $Q^\pi(s,a)$ and $V^\pi(s)$ are the $\alpha$-max-ent action-value and value function respectively of $\pi$.*

For brevity we denote $Q^{\max} \equiv \max_i Q^i$. Define the soft Bellman operator associated with policy $\pi$ as

$$\mathcal{T}^\pi Q(s,a) \equiv r(s,a,s') + \gamma \mathbb{E}_{p(s'|s,a)} \left[\alpha H[\pi(\cdot|s')] + \mathbb{E}_{a'\sim\pi(\cdot|s')}[Q(s',a')]\right].$$

Haarnoja et al. (2018b) have pointed out that the soft Bellman operator $\mathcal{T}^\pi$ corresponds to a conventional, "hard", Bellman operator defined over the same MDP but with reward $r_\pi(s,a,s') = r(s,a,s') + \gamma\alpha\mathbb{E}_{p(s'|s,a)}[H[\pi(\cdot|s')]]$. Thus, as long as $r(s,a,s')$ and $H[\pi(\cdot|s')]$ are bounded, $\mathcal{T}^\pi$ is a contraction with $Q^\pi$ as its fixed point. Appplying $\mathcal{T}^\pi$ to $Q^{\max}(s,a)$ we have:

$$\mathcal{T}^\pi Q^{\max}(s,a) = r(s,a,s') + \gamma\mathbb{E}_{s'\sim p(\cdot|s,a), a'\sim\pi(\cdot|s')}\left[-\alpha\log\pi(a'|s') + Q^{\max}(s',a')\right]$$

$$= r(s,a,s') + \gamma\mathbb{E}_{s'\sim p(\cdot|s,a), a'\sim\pi(\cdot|s')}\left[-\alpha\log\frac{\exp(\alpha^{-1}Q^{\max}(s',a'))}{Z^\pi(s')} + Q^{\max}(s',a')\right]$$

$$= r(s,a,s') + \gamma\mathbb{E}_{s'\sim p(\cdot|s,a)}\left[\alpha\log Z^\pi(s')\right].$$

Similarly, if we apply $\mathcal{T}^{\pi_i}$, the soft Bellman operator induced by policy $\pi_i$, to $Q^{\max}(s,a)$, we obtain:

$$\mathcal{T}^{\pi_i} Q^{\max}(s,a) = r(s,a,s') + \gamma\mathbb{E}_{s'\sim p(\cdot|s,a), a'\sim\pi_i(\cdot|s')}\left[-\alpha\log\pi_i(a'|s') + Q^{\max}(s',a')\right].$$

We now note that the Kullback-Leibler divergence between $\pi_i$ and $\pi$ can be written as

$$D_{\mathrm{KL}}(\pi_i(\cdot|s)\|\pi(\cdot|s)) = \mathbb{E}_{a\sim\pi_i(\cdot|s)}\left[\log\pi_i(a|s) - \log\pi(a|s)\right]$$

$$= \mathbb{E}_{a\sim\pi_i(\cdot|s)}\left[\log\pi_i(a|s) - \frac{1}{\alpha}Q^{\max}(s,a) + \log Z^\pi(s)\right].$$

The quantity above, which is always nonnegative, will be useful in the subsequent derivations. Next we write

$$\mathcal{T}^\pi Q^{\max}(s,a) - \mathcal{T}^{\pi_i} Q^{\max}(s,a) = \gamma\mathbb{E}_{s'\sim p(\cdot|s,a)}\left[\alpha\log Z^\pi(s') - \mathbb{E}_{a'\sim\pi_i(\cdot|s')}[-\alpha\log\pi_i(a'|s') + Q^{\max}(s',a')]\right]$$

$$= \gamma\mathbb{E}_{s'\sim p(\cdot|s,a)}\left[\mathbb{E}_{a'\sim\pi_i(\cdot|s')}[\alpha\log Z^\pi(s') + \alpha\log\pi_i(a'|s') - Q^{\max}(s',a')]\right]$$

$$= \gamma\mathbb{E}_{s'\sim p(\cdot|s,a)}\left[\alpha D_{\mathrm{KL}}(\pi_i(\cdot|s')\|\pi(\cdot|s'))\right]$$

$$\geq 0. \tag{20}$$

From (20) we have that

$$\mathcal{T}^\pi Q^{\max}(s,a) \geq \mathcal{T}^{\pi_i} Q^{\max}(s,a) \geq \mathcal{T}^{\pi_i} Q^i(s,a) = Q^i(s,a) \text{ for all } i.$$

Using the contraction and monotonicity of the soft Bellman operator $\mathcal{T}^\pi$ we have

$$Q^\pi(s,a) = \lim_{k\to\infty} (\mathcal{T}^\pi)^k Q^{\max}(s,a) \geq Q^i(s,a) \text{ for all } i.$$

We have just showed (9). In order to show (10), we note that

$$V^\pi(s) \equiv \alpha H[\pi(\cdot|s)] + \mathbb{E}_{a\sim\pi}[Q^\pi(s,a)]$$

$$\geq \alpha H[\pi(\cdot|s)] + \mathbb{E}_{a\sim\pi}[Q^{\max}(s,a)]$$

$$= \alpha\log Z^\pi(s). \tag{21}$$

Similarly, we have, for all $i$,

$$
\begin{aligned}
V^i(s) &= \mathbb{E}_{a \sim \pi_i(\cdot|s)} \left[ Q^i(s,a) - \alpha \log \pi_i(a|s) \right] \\
&\leq \mathbb{E}_{a \sim \pi_i(\cdot|s)} \left[ Q^{\max}(s,a) - \alpha \log \pi_i(a|s) \right] \\
&= \alpha \log Z^\pi(s) - \alpha D_{\mathrm{KL}}(\pi_i(\cdot|s)\|\pi(\cdot|s)) \\
&\leq \alpha \log Z^\pi(s).
\end{aligned}
\tag{22}
$$

The bound (10) follows from (21) and (22).

## A.2 DC PROOF

**Theorem 3.2 (DC Optimality)** *Let $\pi_i, \pi_j$ be $\alpha$ max-ent optimal policies for tasks with rewards $r_i$ and $r_j$ with max-ent action-value functions $Q^i, Q^j$. Define $C_b^\infty(s_t, a_t)$ as the fixed point of*

$$
C_b^{(k+1)}(s_t, a_t) = -\alpha\gamma \mathbb{E}_{p(s_{t+1}|s_t,a_t)} \left[ \log \int_{\mathcal{A}} \pi_i(a_{t+1}|s_{t+1})^b \pi_j(a_{t+1}|s_{t+1})^{(1-b)} \exp(-\tfrac{1}{\alpha} C_b^{(k)}(s_{t+1}, a_{t+1})) da_{t+1} \right]
$$

*Given the conditions for Soft Q convergence, the max-ent optimal $Q_b^*(s,a)$ for $r_b = br_i + (1-b)r_j$ is*

$$
Q_b^*(s,a) = bQ^i(s,a) + (1-b)Q^j(s,a) - C_b^\infty(s,a) \ \forall s \in \mathcal{S}, a \in \mathcal{A}, b \in [0,1].
$$

We follow a similar approach to Haarnoja et al. (2018a) but without making approximations and generalizing to all convex combinations.

First note that since $\pi_i$ and $\pi_j$ are optimal then $\pi_i(a|s) = \exp(\tfrac{1}{\alpha}(Q^i(s,a) - V^i(s)))$.

For brevity we use $s$ and $s'$ notation rather than writing the time index.

Define

$$
Q_b^{(0)}(s,a) \equiv bQ^i(s,a) + (1-b)Q^j(s,a) \tag{23}
$$

$$
C^{(0)}(s,a) \equiv 0 \tag{24}
$$

and consider soft Q-iteration on $r_b$ starting from $Q_b^{(0)}$. We prove, inductively, that at each iteration $Q_b^{(k+1)} = bQ^i(s,a) + (1-b)Q^j(s,a) - C^{(k+1)}(s,a)$.

This is true by definition for $k = 0$.

$$
Q_b^{(k+1)}(s,a) = r_b(s,a) + \gamma\alpha \mathbb{E}_{p(s'|s,a)} \left[ \log \int_{\mathcal{A}} \exp \frac{1}{\alpha} Q_b^{(k)}(s',a') da' \right] \tag{25}
$$

$$
= r_b(s,a) + \tag{26}
$$

$$
\gamma\alpha \mathbb{E}_{p(s'|s,a)} \left[ \log \int_{\mathcal{A}} \exp(\frac{1}{\alpha}(bQ^i(s',a') + (1-b)Q^j(s',a') - C^{(k)}(s',a'))) da' \right]
$$

$$
= r_b(s,a) + \tag{27}
$$

$$
\mathbb{E}_{p(s'|s,a)} \left[ bV^i(s') + (1-b)V^j(s') + \alpha \log \int_{\mathcal{A}} \exp(b\log\pi_i(a'|s') + (1-b)\log\pi_j(a'|s') - \frac{1}{\alpha}C^{(k)}(s',a')) da' \right]
$$

$$
= bQ^i(s,a) + (1-b)Q^j(s,a) + \tag{28}
$$

$$
\alpha\gamma \mathbb{E}_{p(s'|s,a)} \left[ \log \int_{\mathcal{A}} \exp(b\log\pi_1(a'|s') + (1-b)\log\pi_2(a'|s') - \frac{1}{\alpha}C^{(k)}(s',a')) da' \right]
$$

$$
= bQ^i(s,a) + (1-b)Q^i(s,a) - C_b^{(k+1)}(s,a). \tag{29}
$$

Since soft Q-iteration converges to the $\alpha$ max-ent optimal soft $Q$ then equation 31 holds at the limit.

One can get an intuition for $C_b^\infty(s,a)$ by noting that

$$
C_b^{(1)}(s,a) = \gamma\alpha \mathbb{E}_{p(s'|s,a)} \left[ (1-b) \, \mathrm{D}_b \left( \pi_1(\cdot|s)\|\pi_2(\cdot|s) \right) \right] \tag{30}
$$

where $\mathrm{D}_b$ is the Rényi divergence of order $b$. $C_b^\infty(s,a)$ can be seen as the discount sum of divergences, weighted by the unnormalized product distribution $\pi_1(a|s)^b \pi_2(a|s)^{1-b}$.

### A.3   N POLICIES

It is possible to extend Theorem 3.2 to the case with $N$ policies in a straightforward way.

**Theorem A.1 (Multi-policy DC Optimality)** *Let $\pi_1, \pi_2, ..., \pi_N$ be $\alpha$ max-ent optimal policies for tasks with rewards $r_1, r_2, ..., r_N$ with max-ent action-value functions $Q^1, Q^2, ..., Q^N$.*

*Define $C_{\mathbf{w}}^{\infty}(s_t, a_t)$ as the fixed point of*

$$C_{\mathbf{w}}^{(k+1)}(s_t, a_t) = -\alpha\gamma\mathbb{E}_{p(s_{t+1}|s_t,a_t)}\left[\log \int_{\mathcal{A}} \left(\prod_{i=1}^{N} \pi_i(a_{t+1}|s_{t+1})^{w_i}\right) \exp(-\tfrac{1}{\alpha}C_{\mathbf{w}}^{(k)}(s_{t+1}, a_{t+1}))da_{t+1}\right]$$

*Given the conditions for Soft Q convergence, the max-ent optimal $Q_{\mathbf{w}}^{*}(s,a)$ for and convex combination of rewards $r_{\mathbf{w}} = \sum_{i=1}^{N} r_i w_i$ is*

$$Q_{\mathbf{w}}^{*}(s,a) = \sum_{i=1}^{N} w_i Q^i(s,a) - C_{\mathbf{w}}^{\infty}(s,a)$$

$$\forall s \in \mathcal{S}, a \in \mathcal{A}, \mathbf{w} \in \{\mathbf{w}| \sum_{i=1}^{N} w_i = 1 \ \text{ and } \ w_i \geq 0\}$$

Note that $w_i$ refers to component $i$ of the vector $\mathbf{w}_i$.

The proof is very similar to the two reward case above.

Define

$$Q_{\mathbf{w}}^{(0)} \equiv \sum_{i=1}^{N} w_i Q^i(s,a) \tag{31}$$

$$C_{\mathbf{w}}^{(0)} \equiv 0 \tag{32}$$

and again consider soft Q-iteration on $r_{\mathbf{w}}$. We prove by induction that at each iteration

$$Q_{\mathbf{w}}^{(k+1)}(s,a) = \sum_{i=1}^{N} w_i Q^i(s,a) - C_{\mathbf{w}}^{(k+1)}(s,a) \tag{33}$$

Again, this is true by definition for $k = 0$. Now we consider a step of Soft Q iteration

$$Q_{\mathbf{w}}^{(k+1)} = r_{\mathbf{w}}(s,a) + \gamma\alpha\mathbb{E}_{p(s'|s,a)}\left[\log \int_{\mathcal{A}} \exp \frac{1}{\alpha}Q_{\mathbf{w}}^{(k)}(s',a')da'\right] \tag{34}$$

$$= r_{\mathbf{w}}(s,a) + \gamma\alpha\mathbb{E}_{p(s'|s,a)}\left[\log \int_{\mathcal{A}} \exp \frac{1}{\alpha}\left(\sum_{i=1}^{N} w_i Q^i(s',a') - C_{\mathbf{w}}^{(k)}(s,a)\right)da'\right] \tag{35}$$

$$= r_{\mathbf{w}}(s,a) + \gamma\mathbb{E}_{p(s'|s,a)}\left[\sum_{i=1}^{N} w_i V^i(s') + \alpha\log \int_{\mathcal{A}} \exp\left(\sum_{i=1}^{N} w_i \log \pi_i(a'|s') - \frac{1}{\alpha}C_{\mathbf{w}}^{(k)}(s',a')\right)da'\right] \tag{36}$$

$$= \sum_{i=1}^{N} w_i Q^i(s,a) + \alpha\gamma\mathbb{E}_{p(s'|s,a)}\left[\log \int_{\mathcal{A}} \exp(\sum_{i=1}^{N} w_i \log \pi_i(a'|s') - \frac{1}{\alpha}C^{(k)}(s',a'))da'\right] \tag{37}$$

$$= \sum_{i=1}^{N} w_i Q^i(s,a) - C_{\mathbf{w}}^{(k+1)}(s,a) \tag{38}$$

Since soft Q-iteration converges to the $\alpha$ max-ent optimal soft $Q$ then $Q_{\mathbf{w}}^{*}(s,a) = \sum_{i=1}^{N} w_i Q^i(s,a) - C_{\mathbf{w}}^{(k+1)}(s,a)$ for all $s \in \mathcal{S}, \ a \in \mathcal{A}$.

Note that, in practise, estimating $C_{\mathbf{w}}^{\infty}$ may be more challenging for larger $N$. For compositions of many policies, GPI may be more practical.

# B  Theoretical properties of the composition methods

| Method | Optimal | Bounded loss | Requires $\phi$ | Requires $f(s,a|b)$ |
|--------|---------|--------------|-----------------|---------------------|
| CO     |         |              |                 |                     |
| CondQ  | ✓       | na           | ✓               | ✓                   |
| GPI    |         | ✓            | ✓               |                     |
| DC     | ✓       | na           |                 | ✓                   |

Table 1: Theoretical properties of different approaches to max-ent transfer. The methods compared are: CO, CondQ, max-ent GPI (over a fixed, finite set of policies), and DC. The columns indicate whether the transfer policy is optimal, the regret of the transfer policy is bounded, whether rewards for all tasks $\phi$ need to be observed simultaneously during training and whether the method requires learning a function conditional on the transfer task $b$, $f(s,a|b)$. DC is the only method that both recovers (in principle) the optimal policy and does not require observing $\phi$ during training.

# C  Algorithm details

## C.1  Transfer algorithm

---
**Algorithm 2** AISBP transfer algorithm
---
Load trained parameters $\theta_Q, \theta_q, \theta_s fQ, \theta_C, \theta_{Q_b}$.
Accept transfer task parameter $b$, transfer $method \in$ CO, GPI, DC, CondQ.
**while** testing **do**
    Importance sample transfer policy $\pi_b(a|s) \propto \exp \frac{1}{\alpha} Q^{method}(s,a)$ with mixture proposal $p_b(a|s)_{\theta_q}$.
**end while**

---

## C.2  All losses and estimators

We use neural networks to parametrize all quantities. For each policy we learn an action-value $Q_{\theta_Q}(s,a)$, value $V_{\theta_V}(s)$ and proposal distribution $q_{\theta_q}(a|s)$. We use target networks for the proposal distribution $q_{\theta'_q}(a|s)$ and value $V_{\theta'_V}(s)$.

Here we enumerate all of the losses and their estimators. We use temporal difference (TD(0)) learning for all the RL losses, so all losses are valid off-policy. We use a replay buffer $R$ and learn by sampling minibatches of SARS tuples of size $B$, we index over the batch dimension with $l$ and use $s'_l$ to denote the state following $s_l$, so the tuple consists of $(s_l, a_l, r_l, s'_l)$. For importance sampled estimators we sample $N$ actions for each state $s_l$ and use $a_{lk}$ to denote sample $k$ for state $l$.

We learn a set of $n$ policies, one for each task in $\mathcal{T}$ indexed by $i$. However, we write the losses for a single policy and drop $i$ for notational simplicity.

### C.2.1  Proposal loss

The proposal loss minimizes the KL divergence between the Boltzmann distribution $\pi(a|s) \propto \exp(\frac{1}{2}Q(s,a))$ and the proposal distribution.

$$\mathcal{L}(\theta_q) = \mathbb{E}_R \left[ \mathbb{E}_{a \sim \pi(\cdot|s)}[\log \pi(a|s_t) - \log q_{\theta_q}(a|s_t)] \right] \tag{39}$$

As described in the text, this loss is estimated using importance sampling with a mixture distribution $p(a|s)$ containing equally weighted components consisting of the target proposal distribution $q_{\theta'_q}(a|s)$ for all policies and the uniform distribution.

$$p(a|s) = \frac{1}{n+1} \left( \frac{1}{V^{\mathcal{A}}} + \sum_{i=1}^{n} q^i_{\theta'_q}(a|s) \right) \tag{40}$$

where $V^{\mathcal{A}}$ is the volume of the action space (which is always bounded in our case).

The proposal loss is estimated using self-normalized importance sampling

$$\mathcal{L}(\theta_q) \approx -\frac{1}{B} \sum_{k=1}^{B} \sum_{l=1}^{N} w_{kl} \log q_{\theta_q}(a|s_t), \tag{41}$$

$$w'_{kl} = \frac{\frac{1}{\alpha}(Q_{\theta_Q}(s_k, a_{kl}))}{p(a_{kl}|s_k)}; \quad w_{kl} = \frac{w_{kl'}}{\sum_{m=1}^{N} w'_{km}}. \tag{42}$$

### C.2.2 VALUE LOSS

The soft value loss is

$$\mathcal{L}(\theta_V) = \mathbb{E}_R \left[ \frac{1}{2}(V_{\theta_V}(s_t) - \alpha \log \int_{\mathcal{A}} \exp(\frac{1}{\alpha}Q_{\theta_Q}(s_t, a))da)^2 \right] \tag{43}$$

We estimate this using importance sampling with the proposal distribution $q_{\theta_q}(a|s)$ which is trying to fit the policy $\pi$.

$$\mathcal{L}(\theta_V) \approx \frac{1}{2B} \sum_{l=1}^{B} (V_{\theta_V}(s_l) - \alpha \log Z)^2 \tag{44}$$

$$Z = \left[ \frac{1}{N} \sum_{k=1}^{N} \frac{\exp(\frac{1}{\alpha}Q_{\theta_Q}(s_l, a_{lk}))}{q_{\theta_q}(a_{lk}|s_l)} \right] \tag{45}$$

### C.2.3 ACTION-VALUE LOSS

The TD(0) loss for $Q_{\theta_Q}$ is

$$\mathcal{L}(\theta_Q) = \mathbb{E}_R \left[ \frac{1}{2}(Q_{\theta_Q}(s_t, a_t) - (r(s_t, a_t, s_{t+1}) + \gamma V_{\theta'_V}(s_{t+1})))^2 \right] \tag{46}$$

This does not require importance sampling to estimate and can be straightforwardly estimated as

$$\mathcal{L}(\theta_Q) \approx \frac{1}{2B} \sum_{l=1}^{B} (Q_{\theta_Q}(s_l, a_l) - (r_l + \gamma V_{\theta'_V}(s')))^2. \tag{47}$$

The action-value is parametrized as an advantage function $Q_{\theta_Q}(s, a) = V_{\theta'_V}(s) + A_{\theta_A}(s, a)$.

### C.2.4 STATE DEPENDENT SUCCESSOR FEATURES LOSS

To facilitate max-ent GPI we learn successor features for each policy, both state-action dependent features $\psi_{\theta_\psi}(s, a)$ and state-dependent $\Upsilon_{\theta_\Upsilon}(s)$. As with value, we use a target network for the state-dependent features $\Upsilon_{\theta'_\Upsilon}(s)$

$$\mathcal{L}(\theta_\Upsilon) = \mathbb{E}_R \left[ \frac{1}{2}(\Upsilon_{\theta_\Upsilon}(s_t) - \mathbb{E}_{a_t \sim \pi(a_t|s_t)}[\psi_{\theta_\psi}(s_t, a_t) + \alpha \mathbf{1}(-Q_{\theta_Q}(s_t, a_t) + \alpha \log Z(s_t))])^2 \right]$$

This loss is estimated using self-normalized importance sampling with proposal $q_{\theta_q}$

$$\mathcal{L}(\theta_\Upsilon) \approx \frac{1}{2B} \sum_{l=1}^{B} \sum_{k=1}^{N} w_{lk} \left[ (\psi_{\theta_\psi}^i(s_l, a_{lk}) - Q_{\theta_Q}^i(s_l, a_{lk}) + \alpha \log Z(s_l))^2 \right], \tag{48}$$

$$w_{lk} \propto \frac{\exp(\frac{1}{\alpha}Q^i(s_l, a_l k))}{q_{\theta_q}^i(a_{lk}|s_l)}. \tag{49}$$

We use the importance sampled estimate of $Z$ from eq 47, rather than the value network which may be lagging the true partition function. As with the value estimate, we use self-normalized importance sampling to avoid the importance weights depending on $\alpha \log Z(s_l)$ (this introduces a bias, but in practise appears to work well).

### C.2.5 State-Action Dependent Successor Features loss

The state-action dependent successor feature loss is

$$\mathcal{L}(\theta_{\psi}) = \mathbb{E}_R \left[ \frac{1}{2} (\psi_{\theta_Q}(s_t, a_t) - (\phi(s_t, a_t, s_{t+1}) + \gamma \Upsilon_{\theta'_{\Upsilon}}(s_{t+1})))^2 \right]. \tag{50}$$

for which we use the following estimator

$$\mathcal{L}(\theta_{\psi^i}) \approx \frac{1}{2B} \sum_{l=1}^{B} (\psi^i_{\theta_{\psi}}(s_l, a_l) - (\phi_l + \gamma \Upsilon_{\theta'_{\Upsilon}}(s'_l)))^2. \tag{51}$$

$\psi_{\theta_{\psi}}$ is parametrized as a "psi-vantage" $\psi_{\theta_{\psi}}(s, a) = \Upsilon_{\theta'_{\Upsilon}}(s) + \psi^A_{\theta_A}(s, a)$.

### C.2.6 DC correction

We learn the divergence correction for each pair of policies $\pi^i(a|s)$, $\pi^($a|s)$. As described in the text, in order to learn $C_{\theta_C}(s, a, b)$ for all $b \in [0, 1]$, we sample $b$. We also use a target network $C_{\theta'_C}(s, a, b)$. The loss is then

$$\mathcal{L}(\theta_C) = \mathbb{E}_{s \sim R, b \sim U(0,1)} [\tfrac{1}{2} (C_{\theta_C}(s, a, b) + \alpha\gamma \mathbb{E}_{p(s'|s,a)} [\log \int_{\mathcal{A}} \exp(b \log \pi_i(a'|s') + \tag{52}$$

$$(1 - b)\pi_j(a'|s') - \frac{1}{\alpha} C_{\theta_{C'}}(s', a', b))da'])^2]. $$

This loss is challenging to estimate, due to the dependence on two policies. We importance sample using a mixture of all proposal distributions uniform $p(a|s)$ (equation 40). We denote the samples of $b \sim \mathcal{U}(0, 1)$ for each batch entry $b_l$. The importance sampled estimator is then

$$\mathcal{L}(\theta_C) \approx \frac{1}{N} \sum_{l=1}^{B} \left( C_{\theta_C}(s_l, a_l, b_l) - \alpha\gamma \log \left[ \frac{1}{N} \sum_{k=1}^{N} \frac{C^{target}_{\theta_C}(s'_l, a'_{lk}, b_l)}{p(a_{lk'}|s_l)} \right] \right)^2, \tag{53}$$

$$C^{target}_{\theta'_C}(s'_l, a'_{lk}, b_l) \equiv \exp(\frac{1}{\alpha}(b_l Q^i_{\theta_Q}(s'_l, a'_{lk}) + (1 - b_l)Q^j_{\theta_Q}(s'_l, a'_{lk}) - C_{\theta'_C}(s'_l, a'_{lk}, b_l)). \tag{54}$$

We parametrized $C_{\theta_C}$ as an advantage function $C_{\theta_C}(s, a, b) = C^A_{\theta_{CA}}(s, a, b) + C^B_{\theta_{CB}}(s, b)$ with an additional loss to constrain this parametrization

$$\mathcal{L}(\theta_B) = \mathbb{E}_{a \sim q(\cdot|s), s \sim R} \left[ \frac{1}{2} (C^A_{\theta_{CA}}(s, a, b))^2 \right] \tag{55}$$

which can be straightforwardly estimated by sampling from $q$

$$\mathcal{L}(\theta_B) \approx \frac{1}{2NB} \sum_{l=1}^{B} \sum_{k=1}^{N} (C^A_{\theta_{CA}}(s_l, a_{lk}, b_l))^2 \tag{56}$$

**CondQ** We also consider, as a control, learning the action-value function conditional on $b$ directly (Schaul et al., 2015), in a similar way to the DC correction. We learn both a conditional value $V_{\theta_{V_b}}(s, b)$ and $Q_{\theta_{Q_b}}(s, a, b)$, again by sampling $b$ uniformly each update.

$$\mathcal{L}(\theta_{V_b}) = \mathbb{E}_{R, b \sim U(0,1)} \left[ \frac{1}{2} (V_{\theta_{V_b}}(s, b) - \alpha \log \int \exp(\frac{1}{\alpha} Q_{\theta_{Q_b}}(s, a, b)))^2 \right], \tag{57}$$

$$\mathcal{L}\theta_Q = \mathbb{E}_{R, b \sim U(0,1)} \left[ \frac{1}{2} (Q_{\theta_{Q_b}}(s, a, b) - (r_b + \gamma V_{\theta_{V_b}}(s', b)))^2 \right], \tag{58}$$

where computing $r_b$ for arbitrary $b$ requires $\phi$ to have been observed.

We estimate Cond-Q with the same importance samples as $C$ from $p(a|s)$ and again sample $b \sim \mathcal{U}(0, 1)$ for each entry in the batch. We use target networks for $V_{\theta'_V}(s, b)$ and parametrize $Q_{\theta_Q}(s, a, b) = V_{\theta'_V}(s, b) + A_{\theta_A}(s, a, b)$.

The conditional value estimator is

$$\mathcal{L}(\theta_V) \approx \frac{1}{2B} \sum_{l=1}^{B} \left( V_{\theta_{V_b}}(s_l, b_l) - \alpha \log \frac{1}{N} \sum_{k=1}^{N} \frac{\exp(\frac{1}{\alpha} Q_{\theta_{Q_b}}(s_l, a_{lk}, b_l))}{p(a_{lk}|s_l)} \right)^2 \tag{59}$$

and action-value estimator is

$$\mathcal{L}(\theta_Q) \approx \frac{1}{2B} \sum_{l=1}^{B} \left( Q_{\theta_{Q_b}}(s_l, a_l, b_l) - (r_b + \gamma V_{\theta'_{V_b}}(s'_l, b_l)) \right)^2 \tag{60}$$

## D   JUSTIFICATION FOR THE DC-CHEAP HEURISTIC

We wish to estimate $C_b^\infty(s, a)$ (defined in Theorem A.1) while avoiding learning a conditional function of $b$. We make two (substantial) assumptions to arrive at this approximation.

Firstly, we assume policies $\pi_i(a|s), \pi_j(a|s)$ are Gaussian

$$\pi_i(a|s) = \exp\left( -\frac{(a - \mu_i(s))^2}{2\sigma(s)^2} \right) \tag{61}$$

and the variance $\sigma(s)$ is the same for both policies given a state (it may vary across states).

Secondly, we assume $C_b^{(k)}(s, a) = C_b^{(k)}(s)$ is independent of action. This is approximately correct when nearby states have similar Rényi divergences between policies.

We make use of a result by Gil et al. (2013) that states that the Rényi divergence of order $b$ for two Gaussians of the same variance is

$$D_b \left( \mathcal{N}(\mu_1, \sigma) \| \mathcal{N}(\mu_2, \sigma) \right) = \frac{1}{2} \frac{b(\mu_1 - \mu_2)^2}{\sigma^2}. \tag{62}$$

We first define

$$G_b(s) \equiv (1 - b) D_b \left( \pi_i(\cdot|s) \| \pi_j(\cdot|s) \right) = -\log \int \pi_i(a|s)^b \pi_j(a|s)^{(1-b)} da. \tag{63}$$

From equation 61

$$G_b(s) = 4b(1 - b) G_{\frac{1}{2}}(s). \tag{64}$$

Given these assumptions we show inductively that $C_b^{(k)}(s, a) = 4b(1 - b) C_{1/2}^{(k)}(s, a) \; \forall k, b \in [0, 1]$.

Since $C_b^{(0)}(s, a) = 0 \; \forall b \in [0, 1], a \in \mathcal{A}, s \in \mathcal{S}$ this is true for $k = 0$. We show it holds inductively

$$C_b^{(k+1)}(s, a) = -\alpha\gamma \mathbb{E}_{p(s'|s,a)} \left[ \log \int_{\mathcal{A}} \pi_i(a'|s')^b \pi_j(a'|s')^{(1-b)} \exp(-\frac{1}{\alpha} C_b^{(k)}(s', a')) da' \right] \tag{65}$$

$$= \gamma \mathbb{E}_{p(s'|s,a)} \left[ \alpha G_b(s') + C_b^{(k)}(s') \right] \tag{66}$$

$$= 4(1 - b) b C_{\frac{1}{2}}^{(k+1)}(s, a). \tag{67}$$

Obviously these assumptions are not justified. However, note that we estimate the true divergence for $C_{1/2}^\infty$, i.e. without any assumptions of Gaussian policies and this heuristic is used to estimate $C_b^\infty$ from $C_{1/2}^\infty$. In practise, we find this heuristic works in many situations where the policies have similar variance, particulary when bounded by GPI.

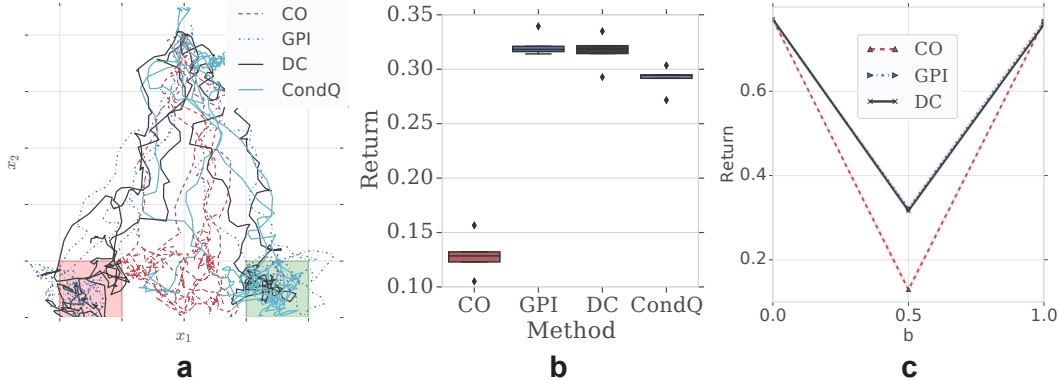

Figure 5: (**a**) Trajectories of the ant during transfer on non-composable subtasks. In this experiment the two base tasks consists of rewards at the red and green square respectively. As expected, in this task, where the two base tasks have no compositional solution, CO (red) performs poorly with trajectories that end up between the two solutions. GPI (blue) performs well, as does DC (black). CondQ does slightly worse.
(**b**) Box-plot of returns from 5 seeds (at $b = 0.5$).
(**c**) Returns as a function of $b$, SEM across 5 seeds is plotted, but is smaller than the line thickness.

## E    ADDITIONAL EXPERIMENT

## F    EXPERIMENT DETAILS

All control tasks were simulated using the MuJoCo physics simulator and constructed using the DM control suite (Tassa et al., 2018) which uses the MuJoCo physics simulator (Todorov et al., 2012).

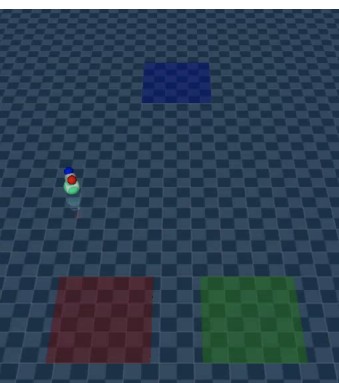

Figure 6: Jumping ball tricky task

The point mass was velocity controlled, all other tasks were torque controlled. The planar manipulator task was based off the planar manipulator in the DM control suite. The reward in all tasks was sparse as described in the main text.

During training for all tasks we start states from the randomly sampled positions and orientations. For the point mass, jumping ball and ant we evaluated transfer starting from the center (in the walker environments, the starting orientation was randomly sampled during transfer, the point mass does not have an orientation). For the planar manipulator transfer was tested from same random distribution as in training. For all tasks were learned infinite time horizon policies.

Transfer is made challenging by the need for good exploration. That was not the focus on this work. We aided exploration in several ways: during training we acted according to a higher-temperature policy $\alpha_e = 2\alpha$. We also sampled actions uniformly in an $\epsilon$-greedy fashion with $\epsilon = 0.1$ and added Gaussian exploration noise during training. This was sufficient to explore the state space for most

tasks. For the planar manipulator and the jumping ball, we found it necessary to induce behavior tasks by learning tasks for reaching the blue target. This behavior policy was, of course, only used for experience and not during transfer.

Below we list the hyper-parameters and networks use for all experiment. The discount $\gamma$ and $\alpha$ were the only sensitive parameters that we needed to vary between tasks to adjust for the differing magnitudes of returns and sensitivity of the action space between bodies. If $\alpha$ is too small then the policies often only find one solution and all transfer approaches behave similarly, while for large $\alpha$ the resulting policies are too stochastic and do not perform well.

| | |
|---|---|
| Proposal learning rate | $10^{-3}$ |
| All other learning rates | $10^{-4}$ |
| Value target update period | 200 |
| Proposal target update period | 200 |
| $\Upsilon$ target update period | 500 |
| Number of importance samples for all estimators during learning | 200 |
| Number of importance samples for acting during training | 50 |
| Number of importance samples for acting during transfer | 1000 |

Table 2: Parameters the same across all experiments

The state vector was preprocessed by a linear projection of $3\times$ its dimension and then a $\mathtt{tanh}$ non-linearity. All action-state networks ($Q$, $\psi$, $C$) consisted of 3 hidden layers with $elu$ non-linearities (Clevert et al., 2015), with both action and preprocessed state projected by linear layers to be of the same dimensionality and used for input the first layer. All value networks and proposal networks consisted of 2 layers with $elu$ non-linearities. The number of neurons in each layer was varied between environments, but was kept the same in all networks and layers (we did not sweep over this parameter, but choose a reasonable number based on our prior on the complexity of the task).

Below we list the per task hyper-parameters

| Task | Number of units | $\alpha$ | $\gamma$ |
|---|---|---|---|
| Point mass | 22 | 1 | 0.99 |
| Planar Manipulator | 192 | 0.05 | 0.99 |
| Jumping Ball | 192 | 0.2 | 0.9 |
| Ant | 252 | 0.1 | 0.95 |

Table 3: Parameters varied between experiments

