# OpenReview forum: "Composing Entropic Policies using Divergence Correction"
_ICLR.cc/2019/Conference_

### Official Review · AnonReviewer3 · 2018-11-03
**Interesting work, but need further improvement**

**Rating:** 5
**Confidence:** 4

**Review:**


-- Contribution, Originality, and Quality --

This paper has presented two approaches for transfer learning in the reinforcement learning (RL) setting: max-ent GPI (Section 3.1) and DC (Section 3.2). The authors have also established some theoretical results for these two approaches (Theorem 3.1 and 3.2), and also demonstrated some experiment results (Section 5).

These two developed approaches are interesting. However, based on existing literature (Barreto et al. 2017; 2018, Haarnoja et al. 2018a), neither of them seems to contain *significant* novelty. The derivations of the theoretical results (Theorem 3.1 and 3.2) are also relatively straightforward. The experiment results in Section 5 are interesting.

-- Clarity --

I have two major complaints about the clarity of this paper.

1) Section 4 of the paper is not well written and is hard to follow.

2) Some notations in the paper are not well defined. For instance

2a) In page 3, the notation \delta has not been defined.
2b) In page 6, both notation V_{\theta'_V} and V'_{\theta_V} have been used. I do not think either of them has been defined.

-- Pros and Cons --

Pros:

1) The proposed approaches and the experiment results are interesting.

Cons:

1) Neither the algorithm design nor the analysis has sufficient novelty, compared to the typical standard of a top-tier conference.

2) The paper is not very well written, especially Section 4.

3) For Theorem 3.2, why not prove a variant of it for the general multi-task case?

4) It would be better to provide the pseudocode of the proposed algorithm in the main body of the paper.

---

> ### Author Response · Authors · 2018-11-17
> **Response to anon reviewer 3**
>
> > ... neither of them seems to contain *significant* novelty. The derivations of the theoretical results
> > (Theorem 3.1 and 3.2) are also relatively straightforward. The experiment results in Section 5 are interesting.
>
> While we acknowledge that our results build on prior work we feel that the reviewer’s assessment undervalues our contributions. To clarify:
>
> Successor Representations/Features have only ever been used in small, discrete action spaces. To the best of our knowledge, we are the first to provide a method for using successor features in continuous action spaces.
> The extension of the GPI theorem to the max-ent RL objective is a non-trivial extension. This brings an important and general idea on value iteration to a new, and important RL framework. It shows that there is a simple, principled way to combine max-ent policies in a way that ensures the result composition at worst retains the performance of the best policy.
>
> The derivation of theorem 3.2 is, as we state in the paper, similar to the approach used in Haarnoja et al., 2018. However, there is an important conceptual leap to show that this term can be practically learned and used to improve performance. Indeed the final lines of Haarnoja are “An interesting avenue for future work would be to further study the implications of this bound on compositionality. For example, can we derive a correction that can be applied to the composed Q-function to reduce bias? Answering such questions would make it more practical to construct new robotic skills out of previously trained building blocks, making it easier to endow robots with large repertoires of behaviors learned via reinforcement learning.” We have taken a first step in this direction and demonstrated that this correction term can be learned in a practical manner.
>
> Additionally, we introduced a very simple heuristic, DC-Cheap. In many cases, this heuristic suffices to get similar performance to DC.
>
> Finally, we also introduce an algorithm for practically using these methods, including, to our knowledge the first example of online zero-shot transfer (in the context defined here) in continuous action spaces. The only other work we are aware of that does this (Haarnoja et al., 2018), requires an offline retraining of the sampler.
>
> >1) Section 4 of the paper is not well written and is hard to follow.
>
> We have re-written section 4. Hopefully it is clearer now. We have also edited the rest of the paper to improve readability. We welcome additional feedback.
>
> > 2a) ... notation \delta has not been defined.
>
> This was the Dirac delta to communicate the GPI policy is deterministic (in this prior work). We have modified this to simplify the notation by explicitly stating the policy is deterministic.
>
> >2b) ... notation V_{\theta'_V} and V'_{\theta_V} have been used. I do not think
> >either of them has been defined.
>
> V_{\theta’_V} is the target network for V_{\theta_V}. This was defined in the paragraph below eq 14. We have now re-phrased this to make it more clear this is the definition. V’_{\theta_V} was a notational mistake, it should have been V_{\theta’_V}. Thank you for pointing that out.
>
> >Pros:
>
> >1) The proposed approaches and the experiment results are interesting.
>
> >Cons:
>
> >1) Neither the algorithm design nor the analysis has sufficient novelty, compared to the typical standard
> > of a top-tier conference.
>
> We addressed this point in detail above, where we enumerated what we believe are the contributions of this work. At the risk of belaboring the point: many papers are simply a more scalable algorithm, or novel theoretical ideas. Here we introduced 2 new theoretical ideas, and a new practical algorithm, is to the best of our knowledge the first demonstration of online zero-shot transfer for task composition in continuous action spaces.
>
> >2) The paper is not very well written, especially Section 4.
>
> We have edited the paper throughout to provide additional clarify and substantially revised section 4 and are happy to make further revisions. Our paper builds upon and extends several lines of work and it has been a challenge to introduce and discuss all relevant concepts in the limited space available.
>
> >3) For Theorem 3.2, why not prove a variant of it for the general multi-task case?
>
> Thank you for the suggestion. We have added a proof in the appendix for the n-policy case of Theorem 3.2. Due to space constraints we have included this in the appendix. he derivation is very similar to the two policy case which we discuss in the main text..
>
> >4) It would be better to provide the pseudocode of the proposed algorithm in the main body of > the paper.
>
> As part of our revision of section 4, we have moved this algorithm into the main text (and modified it to include all the losses.
>
> We thank the reviewer for their time and feedback. We hope we have been able to address many of your concerns and you may reconsider your rating in light of our changes. We welcome additional feedback.

---

### Official Review · AnonReviewer1 · 2018-11-03
**Good Paper**

**Rating:** 7
**Confidence:** 3

**Review:**

This paper proposes using Divergence Correction to compose max ent policies. Based on successor features, this method corrects the optimistic bias of Haarnoja 2018. The motivation for composing policies is sound. This paper addresses the problem statement where policies must accomplish different linear combinations of different reward functions. This method does not require observation the reward weights.

As shown in the experiments, this method outperforms or equally performs past work in both tabular and continuous  environments. The paper is well written and discusses prior work in an informative manner. The tabular examples provide good visualizations of why the methods perform differently.

Minor:
- Figure 1.e: Why does the Optimistic transfer have high regret when the caption says that "on the LU task, optimistic transfers well"
- Figure 1.i states "Neither GPI nor the optimistic policies (j shows GPI, by the Optimistic policy is similar)" but Figure1.j is labeled DC T, is this a typo?
- Figure 2: Many typos:  "(b) Finger position at the en (of the trajectoriesstard ting from randomly sampled start states)"

---

> ### Author Response · Authors · 2018-11-17
> **Response to AnonReviewer1**
>
> We thank the reviewer for their time and feedback. We address the minor corrections below.
>
> >- Figure 1.e: Why does the Optimistic transfer have high regret when the caption says that "on the LU task, optimistic transfers well"
>
> Optimistic transfers much better than GPI on this task. The regret scale is logarithmic, so while Optimistic (red) does not recover the optimal policy, it has a regret approximately 10^4 lower than GPI (blue). We’ve edited the caption to make this clearer. Note in response to another reviewer we have renamed Optimistic to Compositional Optimism (CO) to avoid confusion.
>
> >- Figure 1.i states "Neither GPI nor the optimistic policies (j shows GPI, by the Optimistic policy is similar)" but Figure1.j is labeled DC T, is this a typo?
>
> Apologies, that is indeed a typo. We’ve now reworded this caption.
>
> >- Figure 2: Many typos:  "(b) Finger position at the en (of the trajectoriesstard ting from randomly sampled start states)"
>
> Thank you for pointing this out, this has been fixed.
>
> In general, we made a number of minor edits throughout, and particularly revised section 4 to improve the readability of the paper.

---

> > ### Comment · AnonReviewer1 · 2018-12-12
> > **Response to Response**
> >
> > Thank you for the updates and clarifications. My rating remains the same. I consider this paper to have enough novelty to be interesting. A limiting factor is that ICLR may not be the best venue for this work.

---

### Official Review · AnonReviewer4 · 2018-11-11
**Need clearer motivation for algorithm. Lots of little issues need fixing**

**Rating:** 4
**Confidence:** 3

**Review:**

The authors introduce Divergence Correction (DC) for the problem of transfer learning by composing policies. There approach builds on GPI with a maximum entropy objective. They also prove that DC solves for the max-entropy optimal interpolation between two policies and derive a practical approximation for this algorithm. They provide experimental results in a gridworld problem and study their approximate algorithm in two continuous control problems.

While this paper has some interesting ideas (combining GPI with a Max-Entropy objective and DC), these ideas are not properly motivated. The main problem seems to be clarity. One big problem is that the paper never defines the notion of a notion of optimality (or near-optimality). Also, considering that the DC algorithm is one of the main contributions of the paper it is barely motivated. Theorem 3.2 is presented with almost no explanation about how DC was derived. Why do the authors believe that DC is a good idea on a conceptual level? It's very interesting that the paper presents cases where previous approaches (Optimistic and GPI) don't perform well. But the authors don't explain why they believe DC should perform well in these cases.

The authors make the unjustified claim in the abstract that their approach has "near-optimal performance and requires less information". I say this is unjustified because they only try this approach on three benchmarks. In addition, there should be situations where DC also performs poorly since there are known hardness results for solving MDPs. Admittedly, those results may not apply if the authors are making assumptions that are not being clearly discussed in the paper.

Minor Comments:
1. In the abstract, "requiring less information" is very imprecise. Are you referring to sample complexity?
2. In the introduction, "can consistently achieve good performance" is imprecise. What is the notion of near-optimality? What does consistent mean? Having experimental results on 3 tasks doesn't seem to be enough to me to justify this claim.
3. In the introduction (and rest of the paper), please don't call Haarnoja et al.'s approach optimistic. Optimism already has another widely used meaning in RL literature. Maybe call it "Uncorrected".
4. In section 2.2, the authors introduce \pi_1, \pi_2, ... , \pi_n but never actually use that notation. This section does not clearly explain how GPI works.
5. In Theorem 3.1, the authors should introduce Q^1, Q^2, ... , Q^n and define the policies in terms of the action-value functions. Also, the statement of this theorem is not self contained, what is the reward function of the MDP? The proof below should be called a proof sketch.
6. The paper mentions that extending to multiple tasks is possible. Is it trivial? What is the basic idea? It seems straightforward but it might be helpful to explicitly state the idea.
7. In Theorem 3.2, how was C derived? Please add some commentary explaining the conceptual idea.
8. In Table 1, what is f(s, a|b)? I don't see where this was defined?
9. CondQ is usually referred to as UVFA in the literature.
10. Section 3 really needs a conclusion statement.
11. Section 4 is very unclear and hard to follow.
12. In figure 1f, what is LTD? It's never defined. I'm guessing it's DC.
13. All of the figures are too small and some are not clear in black and white.

---

> ### Author Response · Authors · 2018-11-17
> **Response to AnonReviewer 4 major concerns**
>
> > their approximate algorithm in two continuous control problems.
>
> To clarify: our submission included results for 4 continuous control tasks: 2-D point mass, 5 DOF planar manipulator, 3 DOF jumping ball and 8 DOF ant. We have now added an additional ant task in the appendix (see response later).
>
> > While this paper has some interesting ideas … these ideas are not properly motivated.
> > The main problem seems to be clarity. One big problem is that the paper never
> > defines the notion of a notion of optimality (or near-optimality).
>
> We are interested in 0-shot transfer where we combine existing policies trained on other tasks to provide a solution for a new task. Our notion of optimality is hence with respect to the performance of the optimal policy for the new task. In that sense a policy composition for the transfer task is optimal if it achieved the same performance as the optimal policy for the transfer task. When we say near-optimal, we simply mean the return is nearly that of an optimal policy. We have tried to clarify this notion in our formalism of the transfer problem (section 2.1).
>
> For the tabular tasks we can solve exactly (within numerical precision) for the optimal policy. For the control tasks, we do not have access to the true optimal policy, so we compare the returns of different compositional approaches with one another. We do, however, know roughly the optimal trajectory (i.e. in the “tricky” tasks this corresponds to heading towards the upper right square). We have edited the text to try and clarify these claims.
>
> > … considering that the DC algorithm is one of the main contributions of the paper
> > it is barely motivated.
>
> > Theorem 3.2 is presented with almost no explanation about how DC was derived...
> > why they believe DC should perform well in these cases.
>
> We have added a motivating paragraph before the introduction of DC and have edited this section to provide an intuitive notation of DC before introducing it formally. In short, we want a method that is, in principle (if all components are known exactly) optimal.
>
> Theorem 3.2 is our basis for believing that DC should perform well. It shows that, if all of the terms of Q^{DC} are known exactly, the resulting policy is the optimal policy on the transfer task.
>
> The intuition is that the correction term to (compositional optimism) CO is (roughly) the expected divergences between policies along their trajectories. If the two policies have low-divergences, the CO assumption is approximately correct. If they have high-divergences, this means the policies don’t agree about what actions to take, and thus cannot both achieve their expected returns simultaneously.
>
> > The authors make the unjustified claim in the abstract that their approach has "near-optimal
> > performance and requires less information"...
>
> Firstly, we agree the claim regarding ‘less information’ was ambiguous. The information was not data efficiency (all methods here are trained using the same data). We do not refer here to sample complexity (all methods are trained on the same amount of data, and tested on 0-shot transfer). What is meant by information here is that, under the formalism for transfer used here, GPI and CondQ/UVFA’s require that, while learning policies for each task i, the rewards on all tasks be observed (i.e. \phi is observable). Compositional Optimism and DC do not require access to this information, hence the claim of less information. We have now modified the abstract to explicitly state ``despite not requiring simultaneous observation of all task rewards.’’
>
> As discussed in our response above, we have edited the text to clarify the notion of optimality, which is the standard RL definition (a policy is optimal if it has an expected return the same as the optimal policy for task), but on the compositional transfer task.
>
> The DC theorem, like many RL results, makes the claim of optimality when the components are known exactly. That is, we state in the theorem conditions that $Q^i$, $Q^j$ are the action-value functions of the optimal policies of the base tasks and the correction term C_b^\infty is the solution to the given fixed point. Thus, in some sense, the known hardness results are inside this assumption that we known optimal solutions to the base tasks and the need to know C everywhere. Again, we want to highlight that prior methods do not recover the optimal transfer policy, even when assuming all of their components are known exactly. In the tabular case, where we can (within numerical precision) compute all the components, we do indeed see DC recovers the optimal policy in all tasks we considered.
>
> Practically, our experimental results show that DC does generate better transfer policies than GPI and CO. In these experiments, as with almost all DeepRL, of course we do not have access to the exact action-value’s, but that approximating the DC correction term can result in qualitatively better transfer policies.

---

> > ### Comment · AnonReviewer4 · 2018-12-10
> > **About optimality**
> >
> > Thank you for explaining. I think it would be helpful if you pasted "We are interested in 0-shot transfer where we combine existing policies trained on other tasks to provide a solution for a new task." as the first line of your introduction.
> >
> > The policies that can be composed are fixed {\pi_1, \pi_2, ... , \pi_n}. This means that by optimal policy, you mean the best policy that can be expressed by the value functions of these n policies. This may not be the true optimal value function. This is an important detail that should be clearly defined. For example, if there are m > n actions and each policy corresponds to always choosing one single action. To guarantee that you can recover the true optimal value function, more assumptions are needed about the base set of policies.

---

> > > ### Author Response · Authors · 2018-12-10
> > > **Further clarification on optimality**
> > >
> > > Thank you for your comment and the opportunity for further clarification.
> > >
> > > We cannot change the submission here at the moment but we will clarify the task and are happy to make changes when we are next able to. We currently state that "One approach to this challenge is via the composition of task rewards. Specifically, we are interested in the following question: If we have previously solved a set of tasks with similar transition dynamics but different reward functions, how can we leverage this knowledge to solve new tasks which can be expressed as a convex combination of those rewards functions?" and formalize this in the methods.  We agree that it is a good idea to emphasize this point state this earlier in the manuscript and will incorporate this change.
> > >
> > > When we say optimal, we mean the “true” optimal policy under the maximum entropy objective on the transfer task ... that is the "true optimal policy" under this objective. Indeed, one of the exciting findings of this work is that, with the divergence correction method, it is possible to zero-shot recover the "true" optimal policy under the max-ent objective.
> > >
> > > To understand why this is possible, note that: (1) we use the max-ent objective so that the base policies always assign some likelihood to all actions and, importantly and (2) the transfer tasks are always convex combinations of the base task rewards. The divergence correction term captures exactly the difference between the existing composed value function (Q^CO) and the optimal value function for the transfer task (Q^*). The innovation here is that we do not need any experience on the transfer task to learn the divergence correction, hence we can zero-shot transfer. The divergence corrected policy with this correction term is not simply a product of existing policies, and in some cases takes actions that are unlikely under the base policies (Figure 2 provides an example of this). Appendix A2 contains the formal conditions and proof for this optimality claim (in particular, we assume we know the optimal base policies and action-values along with the correction term C_b). Theorem 3.2 is specifically stating that under these assumptions we CAN guarantee that we recover the true optimal value function during and we have provided a proof there.

---

> ### Author Response · Authors · 2018-11-17
> **Response to AnonReviewer4 minor comments (a)**
>
> Firstly, we apologize for the lengthy response, but we wanted to ensure we addressed all of your comments.
>
> > Minor Comments:
> > 1. In the abstract, "requiring less information" is very imprecise.
> > Are you referring to sample complexity?
>
> This is addressed in the section of the response above to the major concerns.
>
> > 2. In the introduction, "can consistently achieve good performance" is imprecise.
> > What is the notion of near-optimality? What does consistent mean?
> > Having experimental results on 3 tasks doesn't seem to be enough to me to justify this claim.
>
> We have clarified above that we are using the standard notion of optimality in RL, but during the transfer task. We’ve also edited the text to make this clearer. Our claims regarding performance are based on two items: as discussed above, our theoretical results show that DC recovers the optimal policy (if all terms are known exactly). Our empirical results show on 6 (original submission 5) tasks DC performing well.
>
> For our empirical evaluation on continuous control tasks we further focused experiments on the case that emerged as the most difficult in the theoretical analysis and the tabular domains, namely the case when the desired transfer behavior is distinct from that of any of the base policies.
>
> Besides the “tricky” tasks there are two other extremes we considered in the tabular case.
>
> DC method is a correction term to CO, so in the case where CO performs well this implies the correction term to DC is negligible, and we’d expect DC to perform well too (as outlined above, our theoretical results prove DC is optimal on the transfer task with the assumption that all components are exact, so here we mean practical performance with function approximators).
>
> The other extreme is where the two tasks are completely incompatible. In the tabular case, as expected, DC performs well but, one could imagine this task could be challenging for DC in practice, since it implies that the correction term must be large, and potentially challenging to learn. To address this we have added an additional task to the appendix (supplementary figure 5), examining this situation. We find that DC performs as well as GPI, slightly better than CondQ and still much better than CO in this situation.
>
> We use the term “near optimal” in the control tasks to indicate that DC transfer policy trajectories are qualitatively different from the other approaches on the tricky task (i.e. they go towards the optimal joint solution to the task). We’ve now restated this as ``qualitatively better performance,’’ to be more precise, since we don’t have access to the true optimal solution.
>
> In conclusion, Our theoretical analysis demonstrates  that DC recovers the optimal policy during transfer (under the strong assumption that the underlying action-value functions, and of C are known exactly; this analysis is supported by the tabular results). Empirically we have demonstrated that  across the most challenging form of transfers tasks (``tricky’’) in a variety of bodies, DC recover qualitatively different and better policies than other approaches. Finally, our new addition shows in another type of transfer situation (incompatible) tasks, DC performs as well as GPI, and better than CondQ and OC.
>
> > 3. In the introduction (and rest of the paper), please don't call Haarnoja et al.'s
> > approach optimistic. Optimism already has another widely used meaning in
> > RL literature. Maybe call it "Uncorrected".
>
> Thank you for pointing out optimism has another, related, meaning in RL. We do like the term optimism for this approach as it implies the sign of the ‘’uncorrected’ Q (i.e. it always over-estimates the value). For clarity we have now termed this ‘Compositional Optimism’ and used CO throughout, which will hopefully avoid confusion with optimism in the exploration sense.
>
> > 4. In section 2.2, the authors introduce \pi_1, \pi_2, ... , \pi_n but never actually use
> > that notation. This section does not clearly explain how GPI works.
>
> Unfortunately we are limited by the page limit. We have attempted to clarify this somewhat within the space constraints. The set of policies $\pi_1, …$ are used to define the action-values in eq (2). We also try and show in section 3 how we are making use of max-ent GPI in our setup (and of course rely on reference to Barreto et al., for a fuller explanation).
>
> GPI assumes that for a given reward function we have access to the action value functions associated with policies a set pi_1, pi_2, .... In this case we could act according to any of the policies (and would achieve the reward indicated by the associated action value function), but GPI suggests that we can achieve a higher value in all states by acting according to the GPI policy which performs a max over the individual policies' value functions.

---

> ### Author Response · Authors · 2018-11-17
> **Response to AnonReviewer4 minor comments (b)**
>
> > 5. In Theorem 3.1, the authors should introduce Q^1, Q^2, ... , Q^n and define the
> > policies in terms of the action-value functions. Also, the statement of this theorem is not self
> > contained, what is the reward function of the MDP? The proof below should be
> > called a proof sketch.
>
> In this theorem, all policies and action-values Q are for the same MDP (i.e. all action-values are for the same reward function but different policies). In this theorem, the policies are not, in general, defined by their action-values Q (e.g. its not required that the policies are Boltzmann policies). The action-values do need to be the true action-values, so they are defined by their corresponding policies. We’ve tried to add a clarifying note before the theorem (note this statement of the GPI theorem is isomorphic to the standard RL GPI theorem in Barreto et al., 2017). There is no other constraints on the MDP, any valid reward function for an MDP will be valid.
>
> Could you explain why the proof for theorem 3.1 (max-ent GPI) is merely a sketch?. As referenced in the main text, the proof is in appendix A, and 1.5 pages long. Is there a particular step you feel is unclear or missing? We do make use of standard definitions and Soft Q iteration without explanation, but we reference prior work that defines these.
>
> > 6. The paper mentions that extending to multiple tasks is possible. Is it trivial?
> > What is the basic idea? It seems straightforward but it might be helpful to explicitly
> > state the idea.
>
> Max-Ent GPI can be extended to multiple policies straightforwardly as is. The theorem is given for n policies. However, in the original submission we derived the divergence correction only for pairs of base policies.
>
> As suggested by reviewer 2, we have now also added a derivation of the DC term for n policies. Deriving the correction term for more than 2 policies is straightforward, but it may become more difficult to learn for large n. This is one advantage of GPI (no additional complexity is required to make use of n policies) over DC. We had included a discussion of this point in an earlier draft of the paper but we had to remove it due to space constraints. We’ve now added a brief discussion in the appendix where we derive the n-policy DC term.
>
> > 7. In Theorem 3.2, how was C derived? Please add some commentary explaining the
> > conceptual idea.
>
> We address this above. We have re-structured and edited the text to motivate theorem 3.1 prior to introducing it.
>
> > 8. In Table 1, what is f(s, a|b)? I don't see where this was defined?
>
> This notation is only used for the label and is explained in the caption. We’ve modified the caption to explicitly denote this f(s, a|b) to make this clearer. This refers to fact that CondQ and DC both require learning some function that is conditional on b (in DC this is C^\infty_b, in CondQ this is just directly Q(s, a|b).
>
> > 9. CondQ is usually referred to as UVFA in the literature.
>
> We debated whether to refer to this approach as CondQ or UVFA, when we introduce the term CondQ (a contraction of conditional Q) we cite Schaul et al (i.e. the source of the UVFA terminology). However, we felt that many people would understand UVFA’s to refer to a specific architecture (with a dot-product between the task embedding and state embedding to compute value), and the idea of conditioning the value function on a task variable predates UVFAs, so we felt that CondQ was a more neutral  term in this context. We certainly acknowledge UVFAs as a recent demonstration of the scalability of this idea and cite them.
>
> > 10. Section 3 really needs a conclusion statement.
>
> We have added a conclusion statement.
>
> > 11. Section 4 is very unclear and hard to follow.
>
> We agree. We have revised this section substantially. We welcome further of feedback.
>
> > 12. In figure 1f, what is LTD? It's never defined. I'm guessing it's DC.
>
> That is correct. We originally referred to this method as LTD. We have now fixed this. Apologies.
>
> > 13. All of the figures are too small and some are not clear in black and white.
>
> It has been challenging to fit the figures and display all the results concisely. In order allow for discussion we have uploaded a new version of this work, but we will continue to iterate on making the figures clearer.
>
> We thank the reviewer for their time and extensive feedback. We hope we have addressed many of your concerns in this response and revision, and clarified many points of this work. We hope in light of these clarifications you may consider amending your rating.

---

### Author Response · Authors · 2018-11-25
**Please take a second look**

Dear Reviewers,

Thank you for you time looking our submission.

We've tried to respond the issues you raise, including significant improvements to the paper such as rewriting the algorithm section and adding an additional experiment, along with many clarifications.

We'd really appreciate if you could take another look and reevaluate after our changes.

---

### Meta-Review · Area_Chair1 · 2018-12-14
**Multiple reviewers had concerns about the clarity of the presentation and the significance of the results.**

**Confidence:** 3
**Recommendation:** Reject

**Metareview:**

Multiple reviewers had concerns about the clarity of the presentation and the significance of the results.